# Local halide heterogeneity drives surface wrinkling in mixed-halide wide-bandgap perovskites

Kunal Datta [1,2] ✉, Simone C. W. van Laar[1], Margherita Taddei[3], Juanita Hidalgo[2], Tim Kodalle [4,5], Guus J. W. Aalbers [1], Barry Lai[6], Ruipeng Li[7], Nobumichi Tamura [5], Jordi T. W. Frencken [1], Simon V. Quiroz Monnens[1], Robert J. E. Westbrook[3], Daniel J. Graham [8], Carolin M. Sutter-Fella [4], Juan-Pablo Correa-Baena [2], David S. Ginger [3,9], Martijn M. Wienk[1] & René A. J. Janssen [1,10] ✉

Compositional heterogeneity in wide-bandgap (1.8 − 2.1 eV) mixed-halide perovskites is a key bottleneck in the processing of high-quality solution-processed thin films and prevents their application in efficient multijunction solar cells. Notably, mixed-cation (formamidinium-methylammonium) wide-bandgap perovskite films are prone to form micrometer-scale wrinkles which can interfere with the smooth surfaces ideal for multijunction devices. Here, we study the formation dynamics of wrinkled mixed-halide perovskite films and its impact on the local composition and optoelectronic properties. We use in situ X-ray scattering during perovskite film formation to show that crystallization of bromide-rich perovskites precedes that of mixed-halide phases in wrinkled films cast using an antisolvent-based process. Using nanoscopic X-ray fluorescence and hyperspectral photoluminescence imaging, we also demonstrate the formation of iodide- and bromide-rich phases in the wrinkled domains. This intrinsic spatial halide segregation results in an increased local bandgap variation and Urbach energy. Morphological disorder and compositional heterogeneity also aggravate the formation of sub-bandgap electronic defects, reducing photostability and accelerating light-induced segregation of iodide and bromide ions in thin films and solar cells.

Mixed-halide wide-bandgap $ABX_3$ (A is a monovalent cation, B is lead or tin, and X is a halide ion (iodide or bromide)) perovskite semi-conductors are promising candidates for use in monolithic multijunction photovoltaic devices where the use of complementary absorber layers enables an increase in photovoltaic performance[1]. Using compositions with bromide contents of 30%–40% in the top-cell, monolithic tandem solar cells have been developed using perovskite and c-Si bottom sub-cells, which have exceeded 28% and 33% power conversion efficiency respectively[2,3]. Wide-bandgap perovskites with even higher bromide contents (~60%) can

potentially surpass the 40% efficiency mark in monolithic triple-junction devices[4–7].

However, solution-processed mixed-halide perovskites undergo complex crystallization routes that influence their compositional and morphological homogeneity[8–10]. A key challenge is the presence of distinct wrinkled morphological domains in mixed-halide perovskite thin films[11–19]. The formation of these wrinkles, with peak- and valley-like features, has been attributed to compressive stress in the film developed during crystallization[16,20]. The morphological disorder and the residual stress created during film formation then reduce device

performance and stability[21,22]. The extent of wrinkling can be controlled by steering the crystallization rate and interactions with the substrate and solvent environments[14,16,18,23]. Wrinkling is especially challenging in developing efficient perovskite-based multijunction devices where several ancillary thin (often less than 10 nm) films are needed for charge extraction, surface passivation, and as a recombination layer. To ensure conformal coverage and form continuous layers, surface planarity is important[24].

Solution-processed perovskites also suffer from stochastic compositional heterogeneity resulting in local richness or deficiency in ionic species[25–31]. This compositional heterogeneity is particularly crucial in mixed-halide compositions where local variations in halide distribution affect the bandgap[32]. In turn, this variation in local halide concentration may influence the local chemical potential, defect distribution, and charge-carrier dynamics[32,33]. Morphological disorder has previously been associated with compositional heterogeneity in cesium-containing perovskite thin films with bandgaps <1.60 eV[13]. However, the correspondence between morphological disorder and compositional heterogeneity in wide-bandgap compositions has not been investigated extensively. This correlation is especially critical because most wide-bandgap, mixed-halide compositions currently suffer from higher defect densities, resulting in lower radiative recombination efficiencies and a stronger tendency for ion migration[34,35]. As a result, the role of morphological disorder and compositional heterogeneity in determining defect behavior and subsequent effect on photostability also lacks understanding.

Herein, we study perovskite thin films and devices using mixed-cation (formamidinium (FA)–methylammonium (MA)) mixed-halide (iodide–bromide) compositions prepared using an antisolvent-based deposition method. We first identify the role of the [FA]/[MA] and [I]/[Br] ratios in determining morphological disorder and their influence on crystallographic structure and orientation. Thereafter, we study the crystallization dynamics using synchrotron-based in situ structural characterization and identify key stages during processing where such disorder begins to develop. Furthermore, using synchrotron-based nanoscopic X-ray fluorescence mapping, we correlate morphological disorder to the compositional inhomogeneity in local halide distribution. This compositional heterogeneity causes local changes in the bandgap and photoluminescence energy. Finally, using sensitive photocurrent spectroscopy, we associate heterogeneous compositions with a higher defect density and a corresponding instability under illumination due to light-induced halide segregation.

## Results

Mixed-cation lead mixed-halide perovskite (nominal composition $(FA_{1-x}MA_x)Pb(I_{1-y}Br_y)_3$) thin films (compositions denoted as $\{x/y\}$ where $x$ refers to MA content and $y$ refers to Br content) were prepared using an ethyl acetate antisolvent-based spin-coating route as detailed in the "Methods" section. Independently changing the methylammonium ($x$) or bromide ($y$) contents in the precursor solution alters the bandgap between 1.55 eV (25% MA/0% Br) and 2.38 eV (75% MA/100% Br), as determined by ultraviolet-visible-near infrared (UV-vis-NIR) spectroscopy (Fig. 1a and Supplementary Fig. 1)[19]. Calculations indicate that the ideal perovskite bandgap is 1.80–1.90 eV or 1.90–2.00 eV for the top-cell in all-perovskite tandem or perovskite-based triple-junction solar cells, respectively[1]. This bandgap corresponds to a bromide content of 40–60% for the two applications (identified by shaded regions in Fig. 1a).

We used scanning electron microscopy (SEM) to characterize the surface morphology of the resulting perovskite thin films. Figure 1b–g shows low magnification surface SEM images of films with compositions 25% MA/40% Br, 50% MA/40% Br, 75% MA/40% Br, 25% MA/60% Br, 50% MA/60% Br, and 75% MA/60% Br. The images show a smooth surface in films with lower MA and Br contents (25% MA/40% Br and 50% MA/40% Br) whereas other compositions that are richer in MA

and/or Br (75% MA/40% Br, 25% MA/60% Br, 50% MA/60% Br, and 75% MA/60% Br) show additional morphological features on the surface. Specifically, we observe large peaks and valleys on the surface of the films with compositions 75% MA/40% Br, 50% MA/60% Br, and 75% MA/60% Br whereas the film with composition 25% MA/60% Br exhibits a morphology intermediate to the smooth and rough surfaces observed. Similar behavior is observed for films with 50% Br where increasing the MA content increases the presence of such features (Supplementary Fig. 2). Hereafter, we refer to these compositions as $\{x|y\}$ for smooth films, $\{x\vdots y\}$ for rough films, and $\{x\vdots y\}$ for films with intermediate morphology. For example, $\{x/y\} = 0.25|0.40$ forms a smooth film, $\{x/y\} = 0.75\vdots0.60$ forms a rough film, and $\{x/y\} = 0.25\vdots0.60$ has an intermediate behavior. We note that the behavior is also consistent at lower (0%) and higher (100%) MA contents (Supplementary Fig. 3) and that the appearance of morphological disorder does not significantly affect SEM features at 1–2 μm size or create voids in the film surface (Supplementary Fig. 4).

Figure 1h, i shows three-dimensional atomic force microscopy (AFM) profiles of two films with compositions $\{x/y\} = 0.25|0.40$ (smooth), and with higher MA and Br content $\{x/y\} = 0.50\vdots0.60$ (rough) to better visualize the distinct peak- and valley-like features observed in rough films. Figure 1j–o shows line profiles of surface atomic force micrographs (Supplementary Figs. 5 and 6). The feature sizes in the smooth films ($\{x/y\} = 0.25|0.40$, 0.50|0.40, and 0.25|0.60) are on the order of 0.10 μm whereas compositions that yield rough films ($\{x/y\} = 0.75\vdots0.40$, 0.50\vdots0.60, and 0.75\vdots0.60) exhibit feature sizes as large as 1.5–2.0 μm. The line cuts also show that the widths of the features in the rough films are on the order of 2.0–5.0 μm[14,16]. This results in an increase in the average maximum profile height ($R_p$) and root mean square average roughness ($R_q$) as a function of increasing MA and Br contents (Supplementary Fig. 7).

We used synchrotron-based grazing-incidence wide-angle X-ray scattering (GIWAXS) measurements to understand the crystallographic properties of smooth and rough perovskite thin films (Fig. 2). A shift to a higher scattering vector ($q$) for the peak corresponding to the (100) plane (from $q \approx 1.02$ to 1.06 Å$^{-1}$) with increasing MA or Br content suggests the lattice contraction upon the substitution of small cation/anion species (circular averages are shown in Supplementary Fig. 8)[19]. The peak position corresponding to unreacted PbI$_2$ ($q \approx 0.9$ Å$^{-1}$) remains unchanged across compositions, albeit with an increase in preferential orientation with increasing MA content (Supplementary Fig. 9).

Notably, we find that compositions that show roughness >1 μm (Fig. 1) also show a preferential crystallographic orientation. For example, compositions yielding smooth films ($\{x/y\} = 0.25|0.40$, 0.50|0.40, 0.25|0.50, 0.50|0.50, and 0.25|0.60) show GIWAXS patterns (Fig. 2a, b, d, e, g) with broad and complete ring-like features, consistent with randomly oriented crystallites (Fig. 2j). In contrast, films with roughness >1 μm with composition ($\{x/y\} = 0.75\vdots0.40$, 0.75\vdots0.50, 0.50\vdots0.60, and 0.75\vdots0.60) show GIWAXS patterns indicating preferential orientation of the (100) plane due to the appearance of intense spots on the ring at $q \approx 1.0$ Å$^{-1}$ (Fig. 2c, f, h, i), especially in the out-of-plane direction[18]. Here, the inclusion of Br has a stronger influence in driving preferential orientation than MA (Fig. 2j–l). The change in preferential orientation does not arise from compositional changes only. For example, perovskite films of nominally identical compositions deposited using a two-step interdiffusion method in which wrinkling does not occur, show the random orientation of crystallites for compositions with high MA and Br content, as opposed to the preferential orientation observed in films coated using the one-step antisolvent method (Supplementary Fig. 10)[16,18,36].

We then performed synchrotron-based in situ GIWAXS measurements during film formation to characterize the crystallization dynamics of smooth and rough perovskite thin films. For these measurements, we processed the perovskite layer on an indium tin oxide

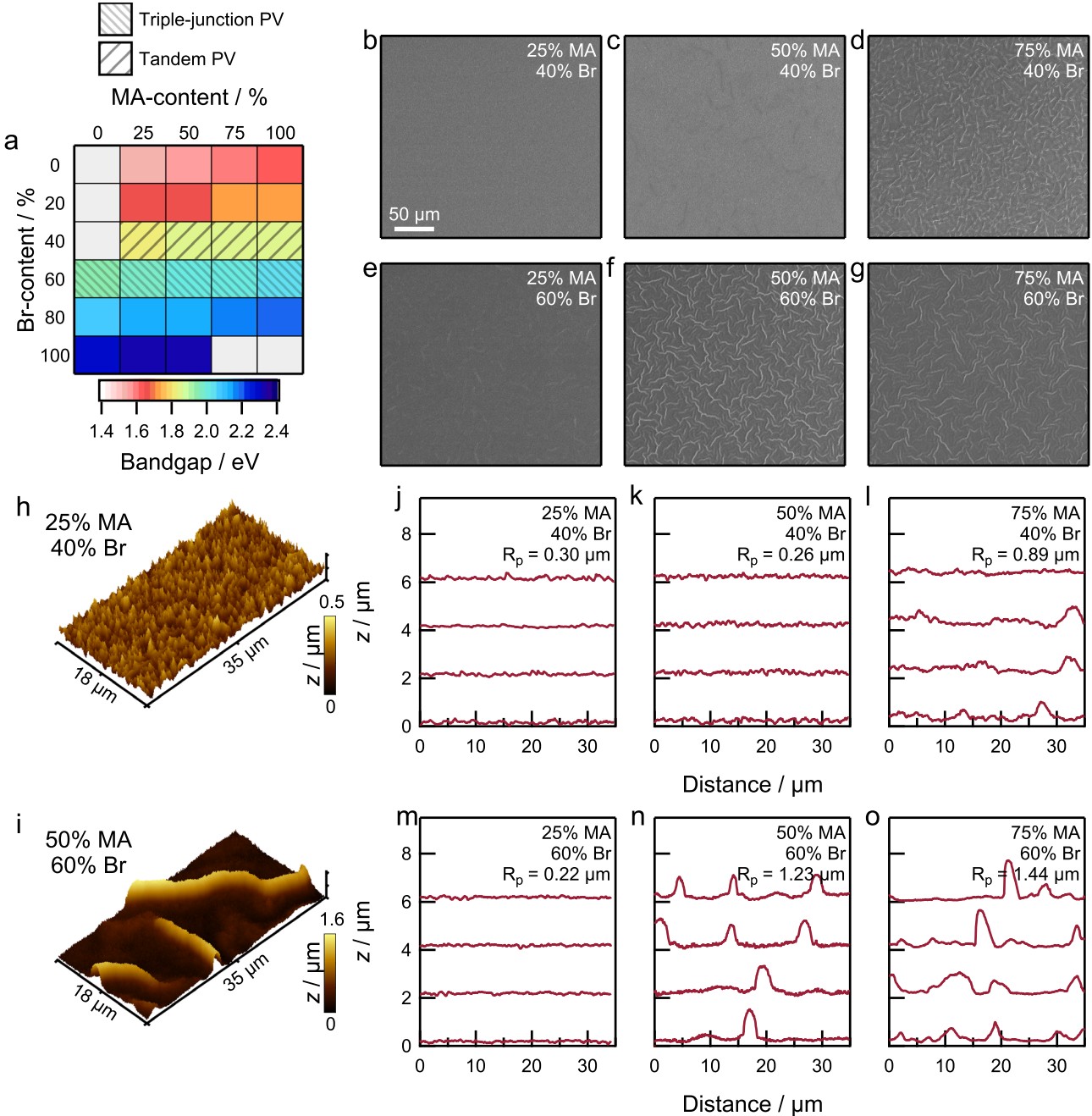

**Fig. 1 | Morphological disorder in mixed-halide perovskite thin films. a** Bandgap from ultraviolet-visible-near infrared (UV-vis-NIR) spectroscopy as a function of $(FA_{1-x}MA_x)Pb(I_{1-y}Br_y)_3$ perovskite composition. Gray areas represent compositions that were not considered due to phase instabilities in ambient measurement environments. Shaded regions mark perovskite compositions compatible for top-cells in tandem and triple-junction solar cells. **b–g** Surface scanning electron microscopy (SEM) images of perovskite thin films for different compositions {x/y}.

**b** {x/y} = 0.25|0.40. **c** {x/y} = 0.50|0.40. **d** {x/y} = 0.75|0.40. **e** {x/y} = 0.25|0.60. **f** {x/y} = 0.50|0.60. **g** {x/y} = 0.75|0.60. **h, i** Three-dimensional atomic force microscopy (AFM) height profiles of perovskite thin films. **h** {x/y} = 0.25|0.40. **i** {x/y} = 0.50|0.60. **j–o** Line cuts from AFM height profiles and average maximum profile height ($R_p$) for perovskite thin films for different compositions {x/y}. **j** {x/y} = 0.25|0.40. **k** {x/y} = 0.50|0.40. **l** {x/y} = 0.75|0.40. **m** {x/y} = 0.25|0.60. **n** {x/y} = 0.50|0.60. **o** {x/y} = 0.75|0.60.

(ITO)-coated glass substrate at a spin-coater placed in the X-ray beam path. We chose the compositions (MA-poor {x/y} = 0.25|0.60 and MA-rich {x/y} = 0.75|0.60) such that they yield different morphological outcomes ({x/y} = 0.25|0.60 yields a smooth film while {x/y} = 0.75|0.60 yields a rough film) but are similar in their nominal Br content. Figure 3 shows in situ GIWAXS patterns as a function of spin-coating time, focusing on the time range near the antisolvent casting at t = 25 s.

Both compositions show no scattering features in the initial stages (up to 21 s, Fig. 3a, b) of the spin-coating process when the wet film is largely disordered. Upon adding the antisolvent (Fig. 3c, d), the film

with composition {x/y} = 0.75|0.60 immediately shows a strong scattering feature in the out-of-plane direction, corresponding to the (100) crystallographic plane of the perovskite phase. In contrast, the (100) feature is absent in the film with composition {x/y} = 0.25|0.60, appearing later at 29 s (Fig. 3e, f) and increasing in intensity thereafter (Fig. 3g, h). Here, as seen in Fig. 2g, the feature resembles a ring which is indicative of a random orientation of crystalline perovskite phases. We also observe similar crystallization dynamics in the characterization of other smooth ({x/y} = 0.25|0.50) and rough ({x/y} = 0.75|0.50) perovskite films (Supplementary Figs. 11 and 12).

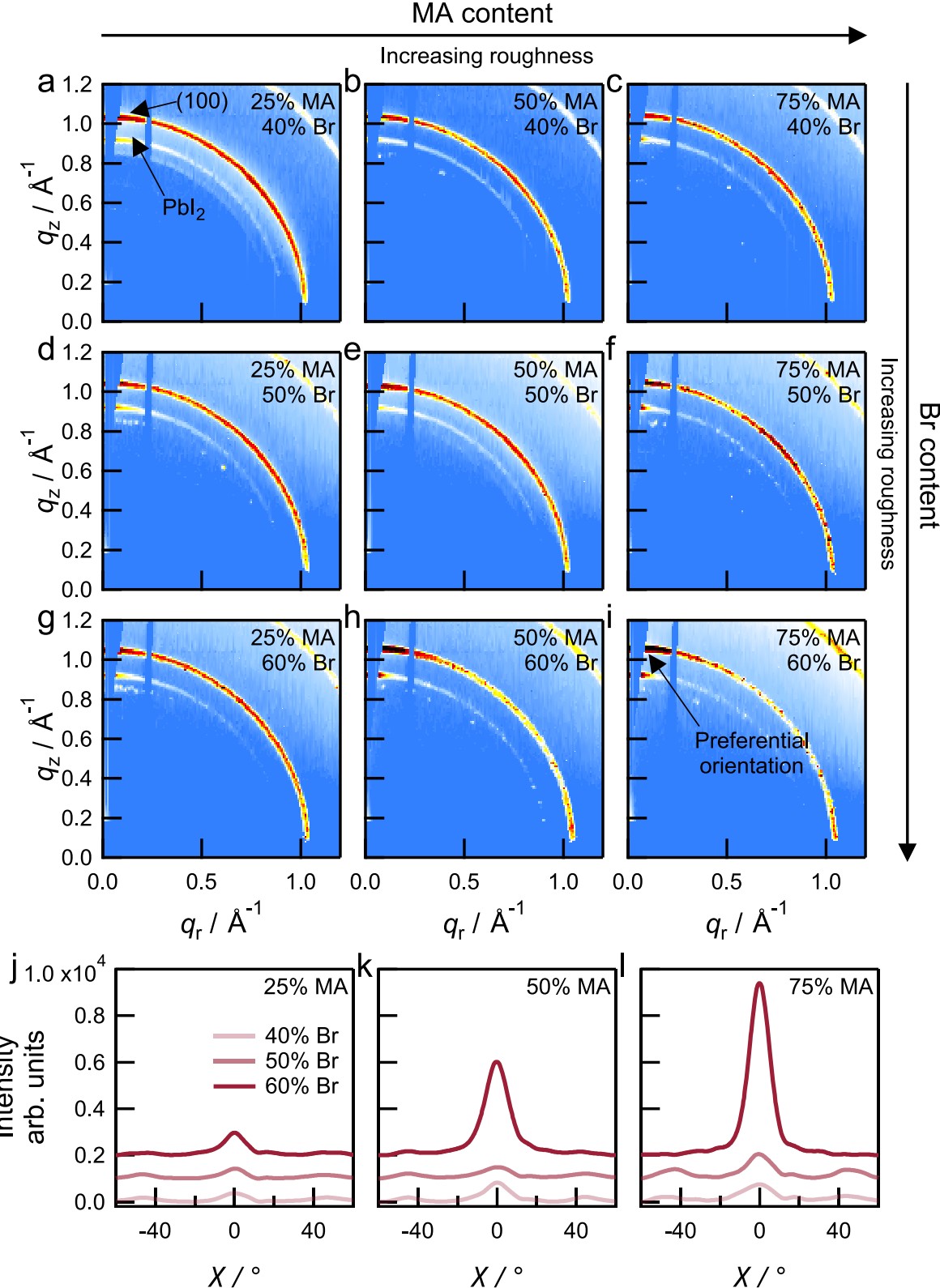

**Fig. 2 | Structural characterization of mixed-halide perovskite thin films.** Grazing-incidence wide-angle X-ray scattering (GIWAXS) patterns of perovskite films with different compositions. **a** {x/y} = 0.25|0.40. **b** {x/y} = 0.50|0.40. **c** {x/y} = 0.75|0.40. **d** {x/y} = 0.25|0.50. **e** {x/y} = 0.50|0.50. **f** {x/y} = 0.75|0.50. **g** {x/y} = 0.25|0.60. **h** {x/y} = 0.50|0.60. **i** {x/y} = 0.75|0.60. The peaks corresponding to unreacted PbI$_2$ ($q \approx 0.9$ Å$^{-1}$) and the (100) plane of the perovskite ($q \approx 1.0$ Å$^{-1}$) are marked in (**a**). Azimuthal intensity profiles of the main Debye-Scherrer ring (100) as a function of $\chi$ angle from GIWAXS for perovskite compositions with varying MA and Br contents. **j** 25% MA. **k** 50% MA. **l** 75% MA. The data in **j**–**l** have been vertically offset for clarity. Smooth films show ring-like features for the (100) plane at $q \approx 1.0$ Å$^{-1}$ whereas films with increasing roughness show intense spots corresponding to preferential orientation.

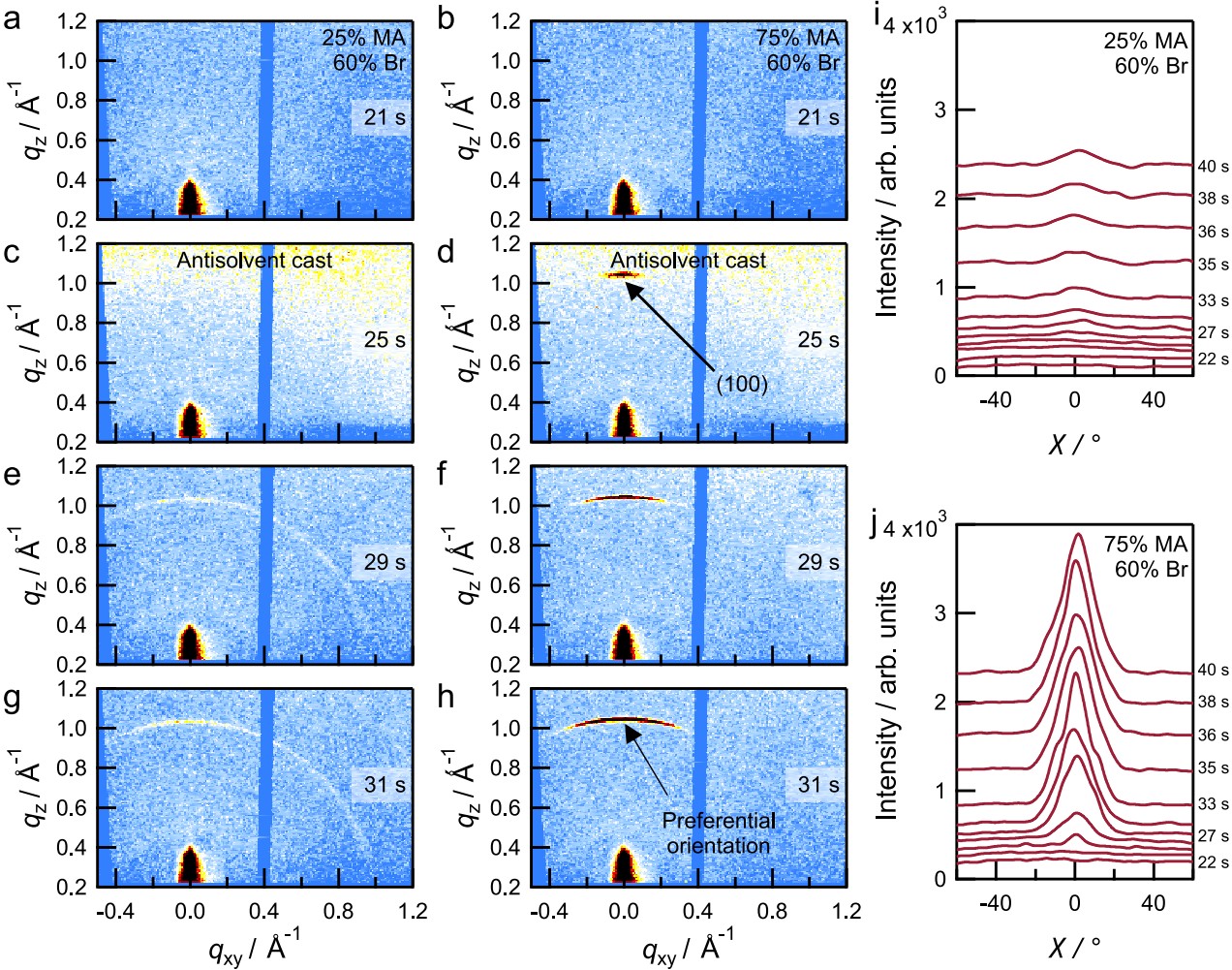

**Fig. 3 | Crystallization dynamics of mixed-halide perovskite thin films.** In situ grazing-incidence wide-angle X-ray scattering (GIWAXS) patterns of perovskites with compositions $\{x/y\} = 0.25|0.60$ and $\{x/y\} = 0.75|0.60$ during spin-coating. Panels mark time stamps during the spin coating process. **a**, **b** 21 s. **c**, **d** 25 s. **e**, **f** 29 s. **g**, **h** 31 s. The frame at 25 s represents the casting of the antisolvent onto the substrate. Azimuthal intensity profiles of the main Debye-Scherrer ring (100) as a function of $\chi$ angle from GIWAXS for different perovskite compositions acquired in the 20–40 s period of spin-coating. **i** $\{x/y\} = 0.25|0.60$. **j** $\{x/y\} = 0.75|0.60$. **a**, **c**, **e**, **g**, **i** refer to $\{x/y\} = 0.25|0.60$ and **b**, **d**, **f**, **h**, **j** refer to $\{x/y\} = 0.75|0.60$. The data in **i** and **j** have been vertically offset for clarity.

These results indicate a faster crystallization and an earlier onset of preferential orientation in the MA-rich $\{x/y\} = 0.75|0.60$ thin film compared to the MA-poor $\{x/y\} = 0.25|0.60$ perovskite[31,37]. This result agrees with prior observations of faster crystallization of Br-rich phases in a MA-based environment than in a FA-based environment[31]. The faster crystallization of Br-rich phases can also be attributed to weak interactions of bromide precursors and DMF, their corresponding poor solubility, and the poor stability of lead bromide complexes in solution, which collectively favor faster nucleation[9,38,39]. Furthermore, it has been demonstrated that Br-based phases form the perovskite phase directly whereas I-based phases undergo crystallization through intermediate complexes[31]. MA-containing Br-based perovskite phases have also been shown to grow as highly oriented phases[37]. As a result, based on the GIWAXS data showing faster crystallization and more preferential (100) orientation in films with roughness >1 μm ($\{x/y\} = 0.75|0.50$ and 0.75|0.60), we propose that in an MA-rich environment, heterogeneous crystal nucleation leads to the formation of oriented bromide-rich perovskites immediately after antisolvent casting (Fig. 3i, j and Supplementary Fig. 13) followed by the incorporation of iodide-containing phases[18], and that the heterogeneous crystallization causes the film wrinkling. The larger change in the

scattering vector ($\Delta q$) for the perovskite composition $\{x/y\} = 0.75|0.60$ during the film-coating and thermal-annealing stages of the crystallization compared to that for $\{x/y\} = 0.25|0.60$, supports the hypothesis that Br- and I-rich phases crystallize at different rates in rough films (Supplementary Fig. 14).

Subsequently, we used synchrotron-based nanoscopic X-ray fluorescence (nano-XRF) microscopy to study the bulk composition of perovskite layers and observe the influence of morphological disorder on compositional heterogeneity (Fig. 4). The technique relies on elemental X-ray emission signatures to map the microscopic distribution of constituents[25,32]. Here, we used the elemental map of Pb (Supplementary Fig. 15) as an indicator for film thickness since peak-like features increase the local film volume that corresponds to a higher X-ray fluorescence intensity. As a result, regions with higher Pb content refer to the peak-like regions of wrinkles whereas lower Pb content corresponds to valley-like regions.

We used nano-XRF mapping to analyze the spatial elemental distribution in smooth and rough perovskite thin films. For example, a film with composition $\{x/y\} = 0.25|0.40$, which yields a uniform surface (Fig. 1b), shows a homogeneous distribution of Pb, I, and Br (Fig. 4a). As a result, the iodide-to-bromide ratio in the film also shows a homogeneous distribution (Fig. 4b). We examined four highlighted regions

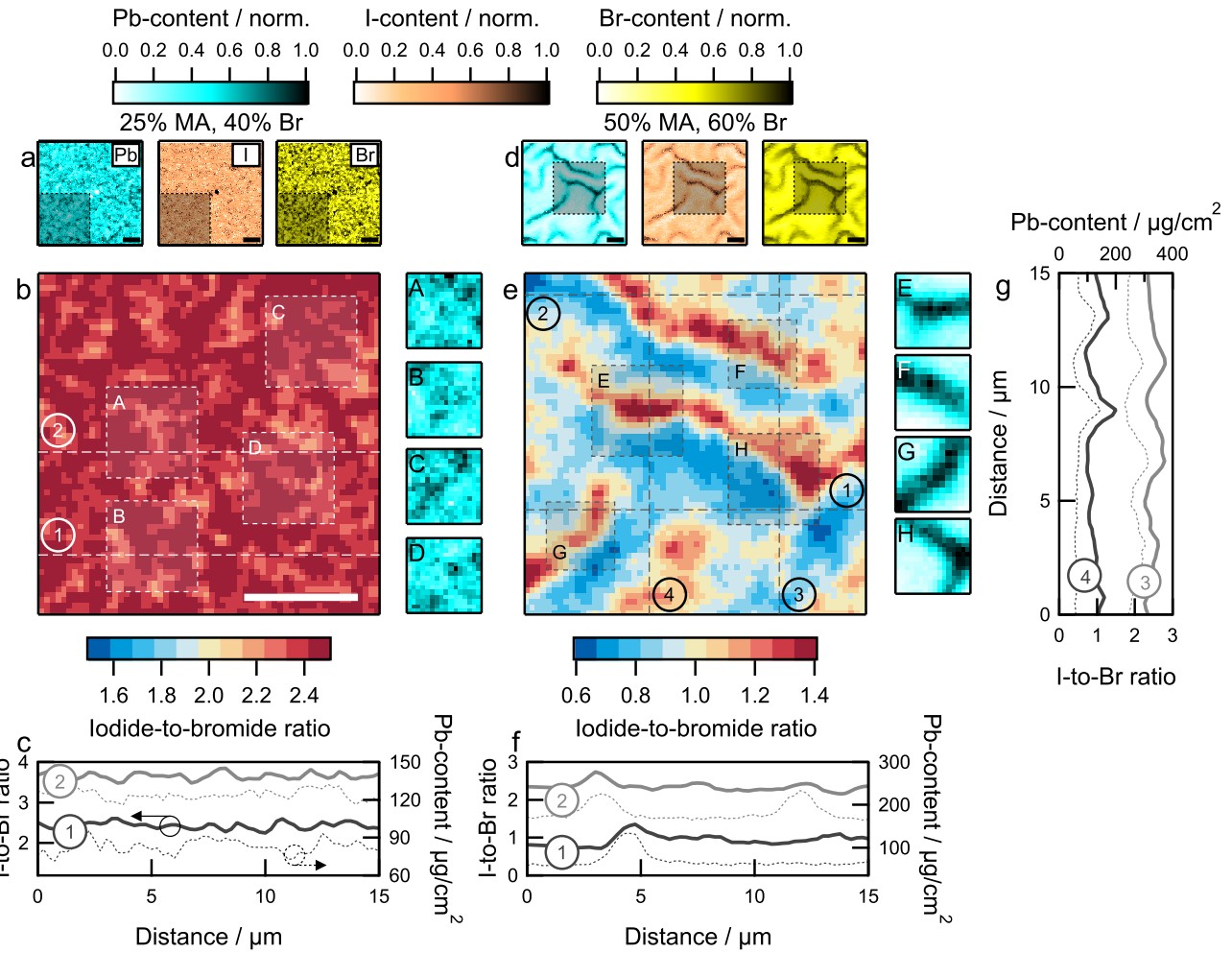

**Fig. 4 | Compositional heterogeneity in mixed-halide perovskite thin films from nanoscopic X-ray fluorescence. a** Normalized elemental maps of Pb, I, and Br of perovskite thin film with {$x/y$} = 0.25|0.40. **b** Map of iodide-to-bromide ratio in shaded region of (**a**). Sub-panels A, B, C, and D represent the normalized Pb elemental map in regions highlighted (dashed squares) in the iodide-to-bromide map. **c** Line cuts of iodide-to-bromide ratios (solid) overlapped with local Pb content line cuts (dashed) marked with (1) and (2) in (**b**). **d** Normalized elemental maps of Pb, I, and Br of perovskite thin film with {$x/y$} = 0.50|0.60. **e** Map of iodide-to-bromide ratio in shaded region of (**d**). Sub-panels E, F, G, and H represent the normalized Pb elemental map in regions highlighted in the iodide-to-bromide map. **f, g** Line cuts of iodide-to-bromide ratios (solid) overlapped with local Pb content line cuts (dashed) at points marked with (1), (2), (3), and (4) in (**e**). Maps show that smooth films yield homogeneous halide distribution across the film thickness and wrinkled films have iodide-rich domains concentrated at peak-like regions. All scale bars are 10 μm.

(dashed boxes in Fig. 4b) to confirm that the iodide-to-bromide ratio is largely independent of the Pb content in the region, i.e., the local thickness of the film. Line cuts of iodide-to-bromide ratio overlapped with the Pb content (Fig. 4c) further confirm the lack of correspondence between the two quantities for a smooth perovskite film. Similar homogeneous distribution of iodide-to-bromide ratio is observed for other smooth films with compositions {$x/y$} = 0.50|0.40, 0.50|0.50, and 0.25|0.60 (Supplementary Fig. 16). We note here that the stochastic distribution of ions causes local nanometer-scale domains to develop that are rich in iodide or bromide ions[26,28].

In contrast, compositions that yield rough surfaces {$x/y$} = 0.75| 0.40, 0.75|0.50, 0.50|0.60, 0.75|0.60, show a heterogeneous distribution of Pb, Br, and I (Fig. 4d and Supplementary Fig. 17). For example, the elemental maps (Pb, I, and Br) for the perovskite film with composition {$x/y$} = 0.50|0.60 (Fig. 4d) show the presence of peak- and valley-like features, as observed in the surface SEM characterization (Fig. 1). Furthermore, the iodide-to-bromide ratio map (Fig. 4e) shows regions of iodide richness and deficiencies which follow the peaks and valleys observed through the Pb elemental map (highlighted regions in Fig. 4d). Line cuts at four distinct locations of the film and two-dimensional maps (Fig. 4f, g and Supplementary Fig. 18) further confirm the positive correlation between the local increase in layer

thickness (Pb content) and higher iodide concentration. Based on these observations, we hypothesize that compositional differences (MA- and Br-richness) influencing the rates of crystallization of iodide- and bromide-rich phases[31,32,38] drive the development of spatial halide heterogeneity during film crystallization[40], with iodide-rich phases forming in the peak-like regions of the films whereas bromide-containing phases predominantly crystallize at the valley-like regions. Compositional heterogeneity has previously been discussed in lead halide perovskites[25,32,41]; however, this work is the first to report its association with surface morphology in wide-bandgap compositions relevant for multijunction devices.

Because of the link between perovskite bandgap and stoichiometry, we expect that compositional heterogeneity will also cause spatial variation in bandgap with regions of high iodide content exhibiting lower bandgaps, and high relative bromide content exhibiting wider bandgaps respectively[17,30,42]. To test this hypothesis, we used hyperspectral photoluminescence (PL) imaging to characterize perovskite thin films' emissive properties. We performed hyperspectral PL mapping of as-prepared films (referred to as "pristine"), excited with a mercury halide lamp in the 350–450 nm range at 130 mW cm$^{-2}$ excitation intensity, with an acquisition time of 1 min. We then continuously illuminated the film for another 5 min (450 nm,

130 mW cm$^{-2}$), and measured again with a 1 min acquisition time (referred to as "illuminated"). Figure 5 displays the PL emission peak wavelength maps from those hyperspectral PL imaging of perovskite films deposited on glass with compositions {$x/y$} = 0.25|0.40, 0.75⋮0.40, and 0.75⋮0.50 (Supplementary Fig. S19), illuminating the top surface (all films were encapsulated in the glovebox with UV-curable glue).

As we demonstrated with AFM and SEM earlier, films with low Br content {$y$ = 0.40} and increasing MA content ($x$) show more pronounced wrinkles (Fig. 1). Using hyperspectral PL imaging, we also observe that wrinkling increases with MA content in low Br films {$y$ = 0.40}. In a smooth film with low MA and Br content ({$x/y$} = 0.25| 0.40), the PL emission wavelength is centered at ~690 nm (Fig. 5a, c, d). The spatially averaged PL spectrum also shows a single peak (Fig. 5c), indicating that the emission is the same across the scanned area. On the other hand, for a rougher film with composition {$x/y$} = 0.75⋮0.40, the PL map and spatially averaged spectrum show two distinct emission peaks, indicative of wide- and narrow-bandgap domains within the scanned region (Fig. 5e, g, h). The emission is primarily dominated by wide-bandgap domains, with a peak maximum at ~685 nm (Fig. 5g). Sparsely distributed, narrow-bandgap domains contribute to the second PL peak maximum at 730–750 nm. Additionally, we note that narrow-bandgap sites ($\lambda_{long}$) are comparatively brighter than wide-bandgap sites ($\lambda_{short}$) (Supplementary Fig. 20). We attribute the increased brightness of the lower bandgap regions to charge-carrier funneling from wider bandgap sites, as proposed previously[43]. This emission heterogeneity is in good agreement with the compositional heterogeneity we observed through nano-XRF mapping (Supplementary Fig. 17), where we found the raised peak-like regions to be comparatively iodide-rich and correspondingly emit at lower energies/ longer wavelengths.

Over the range of compositions studied, we observed that PL heterogeneity in the pristine film increases further with increasing the bromide content (Fig. 5i, k, l), in {$x/y$} = 0.75⋮0.50, with the maximum emission wavelength distribution broadening from 650 to 670 nm (Fig. 5l). The spatially averaged emission spectrum shows a short-(655 nm) and long-wavelength (745 nm) emission contribution, consistent with halide heterogeneity as a result of morphological disorder (Supplementary Fig. 17). However, the redshifted emission at the peak-like regions could also result from self-absorption or optical interference effects that occur at increased layer thickness[44,45]. To rule out optical effects, we used time-of-flight secondary ion mass spectrometry (TOF-SIMS) mapping in the hyperspectral imaging area (Supplementary Fig. 21) and found that stronger long-wavelength emission regions correspond to local increase in the iodide-to-bromide ratio. This correlation is strong evidence that the local redshifts of emission wavelength and increase in PL intensity are due to the high iodide content in peak-like regions. Together, TOF-SIMS and hyperspectral mapping correlation verified that PL wavelength mapping can provide important qualitative information about local compositional heterogeneity in these films.

Subsequently, we continuously illuminated the same samples for 5 min with blue (450 nm) light with an intensity of 130 mW cm$^{-2}$ (Supplementary Fig. 22). This illumination was done in order to cause ion migration that is known to lead to light-induced halide segregation[34]. In Fig. 5b, f, j, we show the spatially resolved PL redshift after 5 min of illumination, calculated from Fig. 5a, e, i and Supplementary Fig. 22. In a compositionally homogeneous, smooth film with composition {$x/y$} = 0.25|0.40, continuous illumination does not affect the PL emission maximum (Fig. 5b, d), but only causes a slight broadening of the PL full width at half-maximum (FWHM) from 35 nm to 45 nm (Fig. 5c). In contrast, films with heterogeneous PL emission prior to illumination ({$x/y$} = 0.75⋮0.40 and 0.75⋮0.50) show a redshift in the maximum emission wavelength by approximately 50–100 nm (Fig. 5f, h, j, l). We note that the change in the emission

wavelength after illumination is lower in the peak-like regions ($\Delta\lambda$ ~ 10 nm) compared to the valley-like regions ($\Delta\lambda$ ~ 50 nm) (Fig. 5f, j). This is because in peak-like areas the iodide-to-bromide ratio is already high before illumination, as we show in Supplementary Fig. 21 via correlated TOF-SIMS. For the perovskite film with composition {$x/y$} = 0.75⋮0.40, the initial emission with two peak features evolves to one broad redshifted emission peak indicating the formation of a broad distribution of emissive iodide-rich species (Fig. 5g). The illuminated {$x/y$} = 0.75⋮0.50 film shows a higher low-energy emission of 745 nm (Fig. 5k), and a weak emission from short-wavelength range of 655 nm.

We performed TOF-SIMS mapping of the perovskite film ({$x/y$} = 0.75⋮0.50) before and after continuous illumination to further understand the effect of ion migration as a function of morphological disorder. By tracking the iodide content before and after illumination, we found that peak-like regions in the film undergo an increase in iodide content on the film surface (Supplementary Fig. 23a). In contrast, a smooth film ({$x/y$} = 0.25|0.50) shows a small and homogeneous increase in iodide content across the film surface upon continuous illumination (Supplementary Fig. 23b), indicating comparatively reduced halide migration as expected from hyperspectral PL imaging (Fig. 5b).

Finally, we fabricated perovskite solar cells (Fig. 6a) in an inverted ($p$-$i$-$n$) device architecture using [2-(9$H$-carbazol-9-yl)ethyl]phosphonic acid (2PACz) and C$_{60}$ as hole- and electron-transport layers, respectively (Supplementary Fig. 24). The transport layers were especially chosen for their compatibility with multijunction device architectures and the conformal deposition of C$_{60}$ with thermal evaporation[46]. Sensitive photocurrent spectroscopy was used to characterize defect dynamics in solar cells. The increase in defect density as a function of Br content has previously been studied[35,38,47], but the role of morphological disorder and compositional heterogeneity on electronic defects of wide-bandgap perovskites is relatively poorly understood. Increasing variations in local bandgap-edge have also been related to non-radiative recombination driven by sub-bandgap defects[32].

For a solar cell with composition {$x/y$} = 0.25|0.40 (Fig. 6b), the external quantum efficiency (EQE), calculated from the photocurrent spectrum, shows a flat above-bandgap (>1.8 eV) EQE profile followed by an exponential drop (Urbach tail) at the bandgap-edge of the active layer. Following that, in the sub-bandgap 0.80–1.60 eV region, a clear EQE contribution of approx. $10^{-7}$–$10^{-6}$ can be observed (at approx. 1.35 eV). A second, less prominent, feature can also be observed at lower energies (at approx. 1.0 eV). Previous work has shown that such sub-bandgap features originate from electronic defects near the perovskite/C$_{60}$ interface[48–50], and that changes in the sub-bandgap photocurrent intensity and photocurrent contribution at the Urbach tail upon prolonged illumination can be associated with light-induced halide segregation[47,51,52].

With increasing MA content, the optical bandgap increases, causing a blueshift in the EQE onset. At $x$ = 0.75, additionally, the sub-bandgap contribution (<1.5 eV) appears less pronounced, manifesting as a broad feature instead of a peak as observed in compositions {$x/y$} = 0.25|0.40 and 0.50|0.40. This broadening is likely related to the increased roughness resulting from morphological disorder in such compositions (Fig. 1), which reduces optical interference by increasing light scattering[48]. The morphological disorder as a function of increasing MA content also manifests as an increase in the Urbach energy, indicating an increase in band-edge energetic disorder[32,53,54].

These effects are much more pronounced upon increasing the bromide content ({$x/y$} = 0.75⋮0.50 and 0.75⋮0.60). Both compositions, for example, show that an additional shoulder appears on the Urbach tail (1.6–2.0 eV), likely indicating the presence of low-energy iodide-rich domains in the pristine device (Fig. 6c)[47,51,52]. These observations agree with the non-uniform halide distribution, as observed using

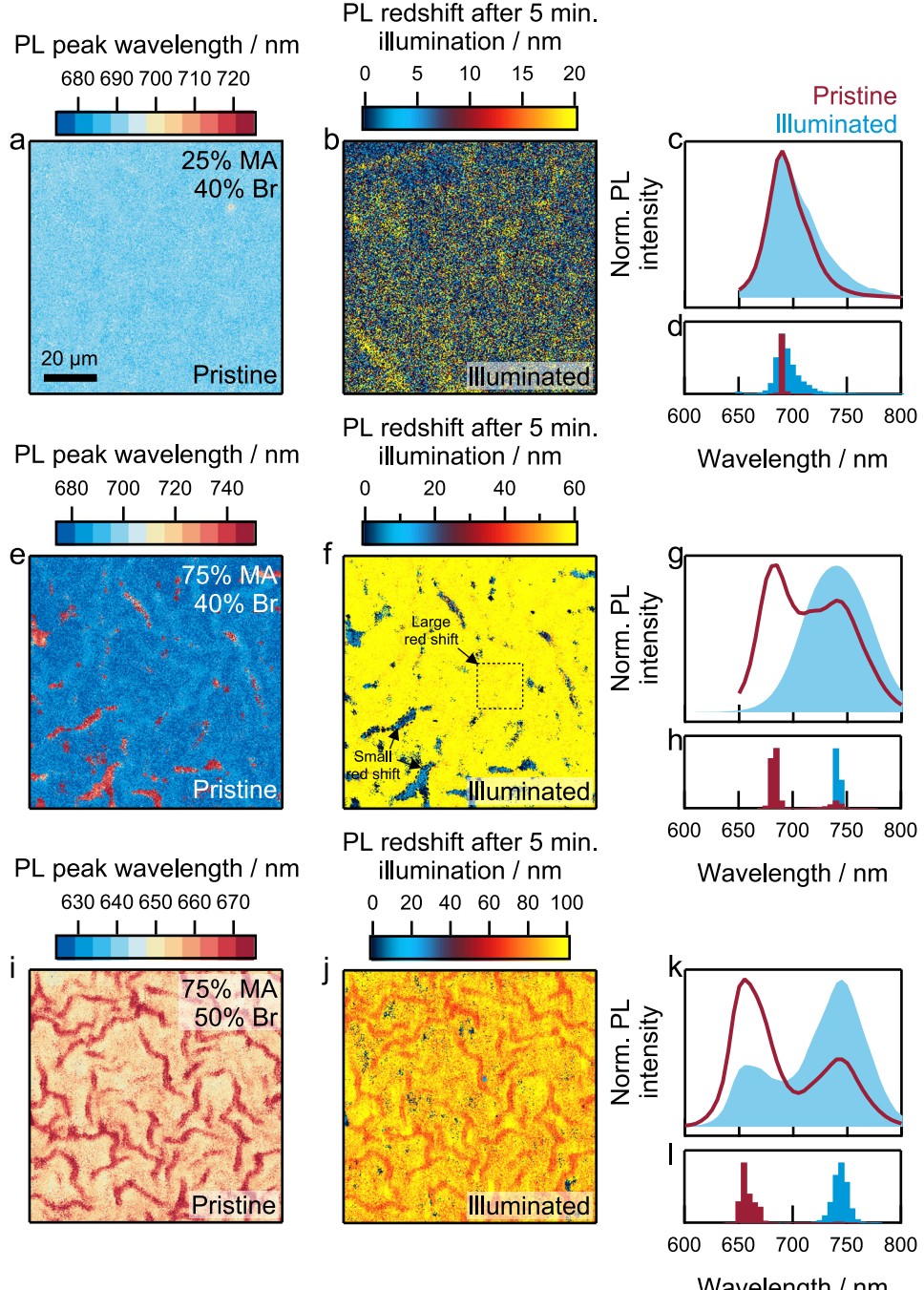

**Fig. 5 | Heterogeneity in photoluminescence emission.** Hyperspectral luminescence of perovskite thin films with compositions. **a**–**d** {x/y} = 0.25|0.40. **e**–**h** {x/y} = 0.75|0.40. **i**–**l** {x/y} = 0.75|0.50. Here, 2D emission maps in **a**, **e**, and **i** represent the wavelength at emission maximum for pristine films. 2D maps in **b**, **f**, and **j** show the wavelength change (Δλ) upon continuous illumination (450 nm, 5 min). Spectra in **c**, **g**, and **k** are averaged over the scanned area of pristine (red line) and illuminated (blue shaded) films. **d**, **h**, and **l** show histogram of maximum emission wavelengths in pristine (red) and illuminated (blue) thin films. The maps show that emission heterogeneity increases with increasing MA and Br content and that in rough films, regions of low-energy emission undergo a smaller redshift after continuous illumination. Note the different color scales in (**a**, **e**, and **i**) and in (**b**, **f**, and **j**).

nano-XRF microscopy (Fig. 4) and hyperspectral PL imaging (Fig. 5). Such an increase in band-edge disorder has been shown to limit the open-circuit voltage in solar cells[55]. Furthermore, the sub-bandgap photocurrent contribution in the 0.8–1.6 eV increases by approx. an order of magnitude for {x/y} = 0.75:0.60 composition, indicating a higher defect density as compared to the composition {x/y} = 0.25| 0.40. The presence of sub-bandgap defects has been consistently associated with high non-radiative recombination in perovskite solar cells[50,56,57]. Taken together, these observations indicate a higher degree

of band-edge disorder and sub-bandgap defect density with increasing compositional/morphological disorder in wide-bandgap perovskites.

We used continuous illumination (532 nm, 1-Sun equivalent intensity, 10 min) to induce defect migration and drive halide segregation in the solar cells[58]. Halide migration in mixed-halide wide-bandgap perovskites can result in redshift of the band-edge, related to the formation of iodide-rich phases, and also cause an increase in the sub-bandgap defect density[34,47,51,52]. Figure 6d–h shows EQE spectra of solar cells following the photo-stress. In solar cells with

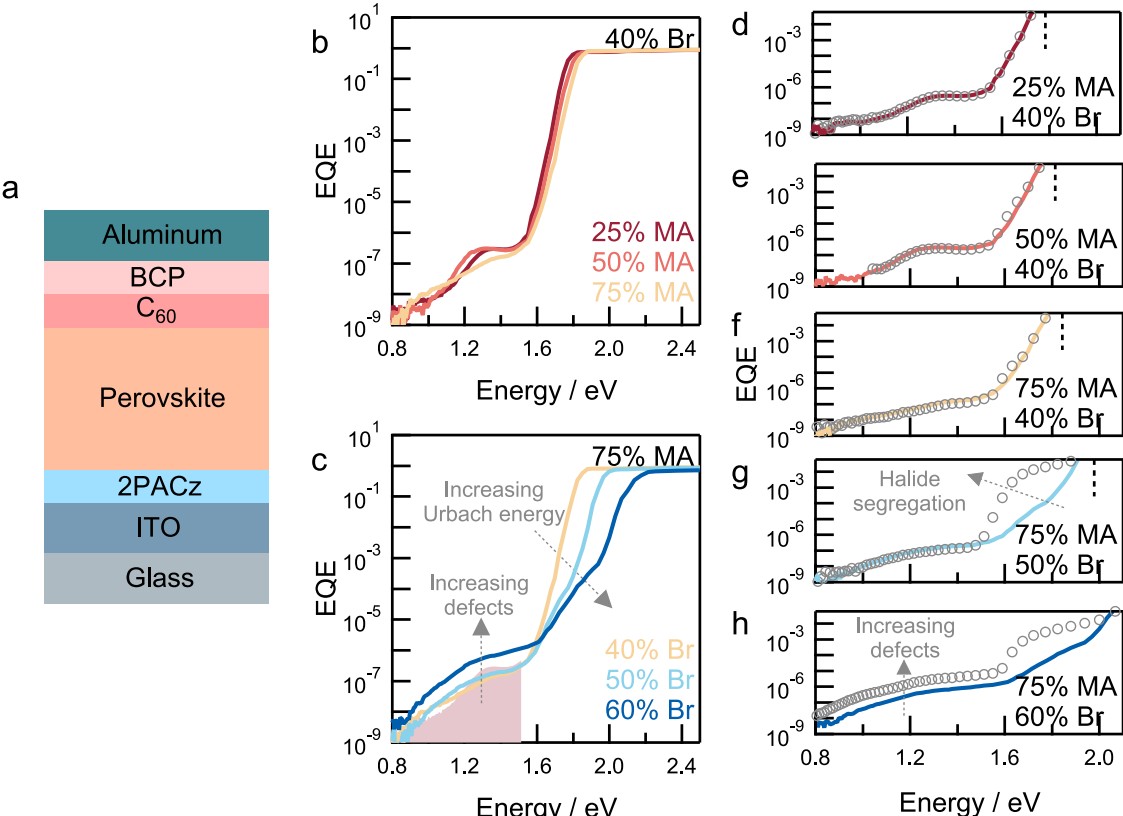

**Fig. 6 | Defect dynamics in mixed-halide perovskite solar cells. a** Device configuration. **b, c** Sensitive external quantum efficiency (EQE) spectra of perovskite solar cells with different absorber layers. **b** Br content ($y$) = 0.40 and MA content ($x$) = 0.25, 0.50, and 0.75. **c** MA content ($x$) = 0.75 and Br content ($y$) = 0.40, 0.50, and 0.60. Shaded region in **c** represents the sub-bandgap EQE spectrum for the $\{x/y\}$ = 0.25|0.40 composition. **d–h** Band-edge and sub-bandgap photocurrent response in the 0.80–2.10 eV range for solar cells using absorber layers for different compositions. Lines represent the spectra of pristine solar cells and open circles represent the spectra acquired after 10 min of continuous illumination (532 nm, 1-Sun equivalent intensity). Dashed vertical lines in **d–h** indicate the respective optical bandgaps. **d** $\{x/y\}$ = 0.25|0.40. **e** $\{x/y\}$ = 0.50|0.40. **f** $\{x/y\}$ = 0.75⋮0.40. **g** $\{x/y\}$ = 0.75⋮0.50. **h** $\{x/y\}$ = 0.75⋮0.60.

low MA and low Br content ($\{x/y\}$ = 0.25|0.40 and 0.50|0.40), that show smooth morphology and minimal compositional heterogeneity (Fig. 6d, e), light-induced changes to the EQE spectrum are minimal, consistent with our observations from hyperspectral PL and TOF-SIMS imaging. In contrast, high MA and high Br containing solar cells with rough perovskite layers ($\{x/y\}$ = 0.75⋮0.40) show a change in the band-edge signal (Fig. 6f), indicating an increase in iodide-rich phase concentration, in agreement with changes observed through PL imaging. Furthermore, in compositions such as $\{x/y\}$ = 0.75⋮0.50 and 0.75⋮0.60 (Fig. 6g, h), a large increase in the band-edge EQE contribution indicates the formation of a large density of iodide-rich phases because of light-induced halide segregation. Moreover, in composition $\{x/y\}$ = 0.75⋮0.60, the change in the Urbach tail is accompanied by an increase in the sub-bandgap defect contribution (0.8–1.6 eV) to the EQE spectrum, indicating an increased defect density upon continuous illumination[47]. Similar trends in photocurrent contribution from band-edge states and sub-bandgap electronic defects are observed in other solar cells using smooth ($\{x/y\}$ = 0.25|0.50) or rough ($\{x/y\}$ = 0.50⋮0.60) perovskite layers (Supplementary Fig. 25).

## Discussion

Controlling morphological disorder and compositional heterogeneity of wide-bandgap perovskite thin films is key to the development of efficient perovskite-based multijunction solar cells. Using the perovskite composition as a variable, we find that MA- and Br-rich compositions are vulnerable to the formation of wrinkled morphology. We use GIWAXS characterization to associate the nominal composition of

the film, the morphology of the layer and its structural properties such as preferential orientation. We identify that at increased MA contents, fast nucleation of Br-rich phases upon antisolvent casting is associated to the development of rough morphology due to heterogeneous crystallization of bromide- and iodide-containing perovskite phases. The resulting features can be 1–2 μm in height and can span 2–5 μm laterally, making them unsuitable for solution-processed multijunction device stacks.

Using nano-XRF microscopy, we find a correspondence between surface wrinkling and halide heterogeneity wherein iodide-rich phases preferentially form at peak-like regions of rough films, influencing the local bandgap distribution across the film. The resulting local narrow-bandgap sites that are rich in iodide ions cause a broadening of band-edges, a higher Urbach energy, and a larger distribution in photoluminescence emission energies, which are potentially detrimental to device performance. Crucially, we also find that compositional heterogeneity and morphological disorder aggravate sub-bandgap defect density. Finally, the combination of high defect density and local halide heterogeneity further reduces material stability to defect-driven light-induced halide segregation under illumination.

Isotropic wavy pattern wrinkles similar to those described in the manuscript have been observed in several thin films, e.g., sol-gel derived zinc oxide and alumina, in $YBa_2Cu_3O_{7−x}$, and perovskites[16,18,59–61]. Their formation has been described theoretically in terms of a bilayer system consisting of an elastic layer, on top of a viscoelastic layer, on top of a rigid substrate[62]. Our findings agree with the proposed mechanism and we rationalize the wrinkle formation as follows. The perovskite precursor solution is initially present as a

homogenous elastic layer. Immediately after the antisolvent is cast a Br-rich perovskite phase crystallizes in the presence of higher MA concentrations, forming a skin layer with a higher elastic modulus than the wet film below. As the remaining more I-rich film below dries and shrinks, it exerts a compressive stress on the top layer. The wrinkling is then a consequence of a spinodal-like decomposition which creates a pattern of ridges and valleys[63]. The valleys are mainly rich in bromide and the ridges are rich in iodide. This entire process sets in within approx. the first 5 s of film crystallization (immediately after antisolvent cast) while the film is still wet. While subsequent annealing promotes crystallization, the orientation and wrinkling has largely occurred already (Supplementary Fig. 13).

The results show how fast nucleation of bromide-rich regions, mediated by the presence of methylammonium, followed by the slower crystallization of iodide-rich regions leads to compositional heterogeneity and morphological disorder that degrades both the performance and stability of the perovskite semiconductor and the resulting solar cells. This insight emphasizes that controlling crystallization kinetics through processing variables such as precursor solubility, film drying conditions and the use of agents that mediate crystallite nucleation and growth can be used to suppress film wrinkling and improve the compositional homogeneity of bromide-rich perovskite compositions[16,18]. We also note previous works that report similar morphological behavior in methylammonium-free perovskite compositions, emphasizing possible limits to charge-carrier dynamics and stability[11,12,14–16,18]. These insights therefore provide potential routes to enhance the performance and stability of wide-bandgap perovskites that currently limit the development of multijunction photovoltaics.

# Methods

## Materials
All materials were used as received and were stored in an inert environment prior to use. FAI, MAI, FABr and MABr were purchased from Greatcell Solar Materials. 2PACz (>98%), PbI$_2$ (99.99%, trace metal basis) and PbBr$_2$ (>98%) were purchased from TCI Chemicals. DMF (99.8%), DMSO (99.9%, anhydrous) and ethyl acetate (99.8%) were purchased from Sigma-Aldrich. C$_{60}$ was purchased from SES Research and bathocuproine (BCP) was purchased from Lumtec. Ethanol (<0.1% H$_2$O) was purchased from Merck Millipore.

## Sample preparation
2PACz was dissolved in anhydrous ethanol at a concentration of 0.33 mg mL$^{-1}$ by sonication prior to use. To prepare the perovskite solution, PbI$_2$ (691.5 mg mL$^{-1}$) and PbBr$_2$ (550.5 mg mL$^{-1}$) were each dissolved overnight at 60 °C in solvent mixtures containing DMF and DMSO in a volumetric ratio of 4:1. The solutions were cooled to room temperature, following which 458 μL PbI$_2$ was added to each 98.4 mg FAI and 91.0 mg MAI respectively. A total of 458 μL PbBr$_2$ was added to each 75.1 mg FABr and 64.1 mg MABr respectively. The resulting solutions (FAPbI$_3$, MAPbI$_3$, FAPbBr$_3$, and MAPbBr$_3$) were stirred at 60 °C for approx. 1 h. Following that, the solutions were mixed based on the desired FA:MA and I:Br ratios and stirred at 60 °C for approx. 1 h. The solutions were then cooled to room temperature prior to use.

Glass and glass/ITO (Naranjo Substrates, 15–17 Ω sq.$^{-1}$) substrates were cleaned by sequential cleaning in acetone, followed by scrubbing with sodium dodecyl sulfate (Acros, 99%) soap solution in deionized water, sonication in soap solution, rinsing in deionized water followed by sonication in 2-propanol. Prior to use, the substrates were exposed to UV-ozone treatment for 30 min after which they were transferred to a N$_2$-filled glove-box.

Perovskite thin films were deposited by spin-coating 150 μL of the precursor at 4000 rpm (5 s to ramp to 4000 rpm) for 35 s. At approx. 25 s from the beginning of the spin-coating, 300 μL of ethyl acetate was cast onto the spinning substrate. The substrates were immediately annealed at 100 °C for 30 min. It must be noted that processing methods are identical across all compositions.

For solar cells, 2PACz was spin-coated at 3000 rpm for 30 s followed by thermal annealing at 100 °C for 10 min. Twenty nm C$_{60}$ and 8 nm BCP were sequentially evaporated at a rate of 0.5 Å s$^{-1}$. A total of 100 nm Ag electrode was thermally evaporated to complete the device. The nominal area of the solar cells determined by the overlap of the ITO and Ag areas is 9 mm$^2$.

## Thin film and solar cell characterization
UV-vis-NIR spectra of perovskite thin films were measured using PerkinElmer Lambda 1050 UV-vis-NIR spectrophotometer. SEM images were acquired using FEI Quanta 3D FEG microscope, operated with a 5 kV electron beam and a secondary electron detector. AFM measurements were conducted using a Dimension 3100 AFM in tapping mode. For determining the $J − V$ characteristics a Keithley 2400 SMU was used. A tungsten halogen lamp, filtered by a Schott GG385 UV filter and a Hoya LB120 daylight filter, was used to simulate 100 mW cm$^{-2}$ of visible light. A shadow mask with 0.0676 cm$^2$ aperture was used to define the illuminated cell area. $J − V$ scans involved sweeping the applied voltage (with no pre-biasing) from +1.5 to −0.5 V at a rate of 0.25 V s$^{-1}$. EQE measurement was performed in a nitrogen atmosphere. The probe light source was generated by a 50 W tungsten-halogen lamp (Philips focusline), which was modulated at 160 Hz with a mechanical chopper (Stanford Research, SR 540) before passing into a monochromator (Oriel, Cornerstone 130). The spectral response of the device was recorded as a voltage from a pre-amplifier (Stanford Research, SR 570) using a lock-in amplifier (Stanford Research, SR 830), and was calibrated by a reference silicon cell. To accurately determine the current density, a green LED (530 nm, Thorlabs M530L3, driven by a DC4104 driver) was used as a light bias to provide the solar cell with approximately one sun illumination intensity.

## GIWAXS characterization
GIWAXS measurements were done at beamline 11-BM at Brookhaven National Laboratory on perovskite films deposited on glass substrates cut to a size of approx. 0.5 cm × 0.5 cm. The samples were measured at an incident angle ($α_i$) 0.5° with a 10 s exposure time. The X-ray beam had an energy of 13.5 keV, 0.2 mm (height) × 0.05 mm (width) size, 1 mrad divergence and an energy resolution of 0.7%. Diffraction images were recorded using a Pilatus 800k detector. Data were analyzed using the SciAnalysis package provided by the beamline. In situ GIWAXS was performed during spin coating and thermal annealing in a custom-made spin-coater attached to beamline 12.3.2 at the Advanced Light Source (ALS), Lawrence Berkeley National Laboratory. The incoming X-ray beam was at an angle of 0.5° with a beam energy of 10 keV. A DECTRIS Pilatus1 M X-ray detector at an angle of 35° to the sample plane and a sample−detector distance of ~186 mm was used. Measurements were carried out on an area of 0.1 mm$^2$ (10 mm × 0.01 mm) under an overpressure of N$_2$ in the spin-coater chamber. Samples were heated using a ramp rate of 4 °C s$^{-1}$. The diffraction data was collected with a framerate of approximately 1.875 s$^{-1}$.

## X-ray fluorescence microscopy
X-ray fluorescence (XRF) microscopy measurements were conducted using the Advanced Photon Source (APS) at beamline 2-ID-D at Argonne National Laboratory. A synchrotron X-ray energy of 14 keV was used and a step size of 0.15 μm and 50 ms dwell time. Data were analyzed using the MAPS software and spectrum fitting was used to deconvolute overlapping peaks and background from fluorescence data. In addition, after a standard calibration, it was possible to quantify the mass concentration in the sample to calculate molar

ratios. The NIST thin-film standards SRM 1832 and 1833 were used for calibration.

## Hyperspectral PL

Hyperspectral measurements were performed using a Photon etc. IMA upright microscope fitted with a transmitted darkfield condenser and a 60X objective (Nikon Plan RT, NA 0.7, CC 0-1.2). The excitation was done using a mercury halide lamp (Nikon ultrahigh pressure 130 W mercury lamp) passing through a 450 nm short-pass filter and emission was collected through a 500 dichroic filter and 550 nm long-pass filter. The lamp has six levels of light intensity, and all the measurements were taken using the lowest intensity (ND32) with total incident power on the sample of 130 mW cm$^{-2}$. The scan duration was 1 min per frame. Post-processing was done in the proprietary Photon etc. PHy-Spec software.

## Time-of-flight secondary ion mass spectrometry

TOF-SIMS negative ion data were acquired on an IONTOF TOF.SIMS5 spectrometer using a 25 keV Bi$^{3+}$ cluster ion source in the delayed extraction mode. The ion source was operated at a current of 0.04 pA that was rastered over a 100 μm × 100 μm area at 256 pixels × 256 pixels for a primary ion dose of $2.5 \times 10^{11}$ ions cm$^{-2}$. A low energy flood gun was used for charge neutralization. Spectra were acquired over a mass range of $m/z = 0$ to 800 amu. The data was calibrated using the C$_4$H-, Br-, I- and PbI$_2$- peaks. Mass resolution of the $m/z = 49.005$ (C$_4$H-) peak was around 3500.

## Sensitive EQE spectroscopy

Sensitive EQE measurements to characterize sub-bandgap states were conducted using an Osram 64655 HLX 250 W halogen lamp as the illumination source. The light was mechanically chopped at 333 Hz using an Oriel 3502 chopper and was subsequently passed through a monochromator (Oriel, Cornerstone 260) and appropriate sorting filters. The solar cell was mounted into a nitrogen-filled sample holder and its response was recorded as a voltage from a pre-amplifier (Stanford Research, SR 570) using a lock-in amplifier (Stanford Research, SR 830). The measurements were calibrated using Si and InGaAs reference cells. To illuminate the solar cells, a 532 nm CW laser (B&W Tek Inc., BWN-532-20E/56486) was used and its intensity and illumination area was adjusted to match 1-Sun equivalent intensity and the solar cell area using a set of neutral density filters and fish-eye lenses. The data was normalized to the drop in photocurrent that marks the band edge in the spectrum of the pristine solar cell. In photostability studies, spectra were scaled for the signal produced from pristine solar cells in order to estimate the change in the above-bandgap EQE.

## Reporting summary

Further information on research design is available in the Nature Portfolio Reporting Summary linked to this article.

## Data availability

Source data are provided with this paper.

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

## Acknowledgements

R.A.J.J. acknowledges the Ministry of Education, Culture, and Science (Gravity program 024.001.035), the Netherlands Organization for Scientific Research (Joint Solar Programme III Project 680.91.011 and Spinoza grant) and the European Research Council (ERC) under the European Union's Horizon Europe research and innovation programme (Grant Agreement No. 101098168, PERSTACK) for funding. M.T. acknowledges the Office of Naval Research (Award # N00014-20-1-2587) for funding. TOF-SIMS was carried out at the Molecular Analysis Facility, a National Nanotechnology Coordinated Infrastructure site at the University of Washington which is supported in part by the National Science Foundation (awards NNCI-2025489, NNCI-1542101), the Molecular Engineering and Sciences Institute, and the Clean Energy Institute. The research used the CMS 11-BM beamline of the National Synchrotron Light Source II, a U.S. Department of Energy (DOE) Office Science User Facility operated for DOE Office of Science by Brookhaven National Laboratory under contract no. DE-SC0012704. The research used beamline 12.3.2, a resource from the Advanced Light Source, a DOE Office of Science User Facility under Contract No. DE-AC02-05CH11231. Work at the Molecular Foundry was supported by the Office of Science, Office of Basic Energy Sciences, of the U.S. Department of Energy under Contract No. DE-AC02-05CH11231. T.K. acknowledges support by the U.S. Department of Energy (DOE), Office of Science, Office of Basic Energy Sciences, Materials Sciences and Engineering Division under Contract No. DE-AC02-05-CH11231 (D2S2 program KCD2S2). This research used resources of the Advanced Photon Source, a U.S. DOE Office of Science User Facility operated for the DOE Office of

Science by Argonne National Laboratory under Contract No. DE-AC02-06CH11357.

## Author contributions

K.D., S.C.W.v.L., M.M.W. and R.A.J.J. planned the research. K.D. and R.L. measured and analyzed GIWAXS. T.K. and N.T. measured and analyzed in situ GIWAXS. J.H., B.L. and K.D. measured and analyzed nano-XRF microscopy. K.D. and S.V.Q.M. performed AFM measurements. M.T. conceived of, measured, and analyzed the hyperspectral PL, with input from D.S.G., and M.T. and D.J.G. performed and analyzed the TOF-SIMS imaging data. S.C.W.v.L., R.J.E.W. and K.D. helped analyze hyperspectral PL microscopy and TOF-SIMS data. G.J.W.A. measured and analyzed sensitive photocurrent spectroscopy. J.T.W.F. assisted K.D. and S.C.W.v.L. in materials development. J.-P.C.-B. helped in the analysis of GIWAXS and XRF data. K.D. wrote the manuscript with input from all co-authors. C.M.S.-F. contributed to the revision of the manuscript.

## Competing interests

The authors declare no competing interests.

## Additional information

[1]Molecular Materials and Nanosystems and Institute of Complex Molecular Systems, Eindhoven University of Technology, P.O. Box 513, 5600 MB Eindhoven, The Netherlands. [2]School of Materials Science and Engineering, Georgia Institute of Technology, Atlanta, GA 30332, USA. [3]Department of Chemistry, University of Washington, Seattle, WA 98195-1700, USA. [4]Molecular Foundry, Lawrence Berkeley National Laboratory, 1 Cyclotron Road, Berkeley, CA 94720, USA. [5]Advanced Light Source, Lawrence Berkeley National Laboratory, 1 Cyclotron Road, Berkeley, CA 94720, USA. [6]Advanced Photon Source, Argonne National Laboratory, Lemont, IL 60439, USA. [7]National Synchrotron Light Source II, Brookhaven National Laboratory, Upton, NY 11973, USA. [8]Department of Bioengineering, University of Washington, Seattle, WA 98195-1653, USA. [9]Physical Sciences Division, Physical and Computational Sciences Directorate, Pacific Northwest National Laboratory, Richland, WA 99352, USA. [10]Dutch Institute of Fundamental Energy Research, De Zaale 20, 5612 AJ Eindhoven, The Netherlands. ✉e-mail: kdatta7@gatech.edu; k.datta@tue.nl; r.a.j.janssen@tue.nl

