## [Transparent Peer Review File · Nature Communications]

Local halide heterogeneity drives surface wrinkling in mixed-halide wide-bandgap perovskites

Corresponding Author: Professor René Janssen

Version 0:

Reviewer comments:

Reviewer #1

(Remarks to the Author)

“Local halide heterogeneity drives surface wrinkling in mixed-halide wide-bandgap perovskites” by Kunal Datta et al. discusses that wrinkling in $(\text{FA}_{1-x}\text{MA}_x)\text{Pb}(\text{I}_{1-y}\text{Br}_y)_3$ perovskite is compositionally driven, requiring high concentrations of MA, and found that the peaks of these wrinkles are iodide-rich while the troughs are bromide-rich. This conclusion is interesting and well-supported by the experimental results. There are a few lingering questions remaining that should be addressed before this manuscript is published, though.

1. One general comment given how many compositions the authors tested was that it was hard to keep straight which compositions were wrinkled and which weren't as I was reading. Perhaps there could be some sort of additional designator in addition to the x/y to help out people like me from having to refer back to figure 1 all the time?

2. It is difficult to tell if the PbI_2 peak is unchanged in the GIWAXS as stated by the authors. There appears to be an orientational change depending on the composition.

3. Also in the GIWAXS, the authors refer to “Fig. 2f, 2h, 2i, 2j and 2l” when discussing intense spots on the ring at $q = 1.0 \text{ \AA}^{-1}$, but it does not seem like all of these figures align with what is written.

4. The correlation between the PL and TOF-SIMS is not presented in a particularly conclusive way. I can maybe see what the authors are stating if I squint, but I could just be seeing what the authors want me to see. Is there a better way to present this information?

5. A key question remaining in this work is are these wrinkles due to film compression and subsequent delamination? If so, what mechanism would be driving volumetric expansion of the film as it is drying? Is it possible that the wrinkles occur where the perovskite is iodide-rich due to the larger size of iodide forcing the perovskite to distort out-of-plane to relax stresses?

Reviewer #2

(Remarks to the Author)

Datta et al. reported an in-depth analysis of compositional heterogeneity in wide bandgap perovskites and its impact on their structural and optoelectronic properties. Authors analyzed different set of samples (varying the A-site cation and halides, respectively) to investigate the wrinkle formation on the surface of the films. The local inhomogeneities (morphological and compositional) were identified as key contributors to film degradation under continuous light exposure. The findings of the significant and timely for research on mixed cation mixed halide perovskites.

1. Page 4: Authors used different FA and MA ratios for A-site cation, and different bromide and iodide ratios for X-site halide effects. In sample labeling, denoting as x/y was hard to follow in the main text. Directly MA/Br percentages would be easier to follow as in figure insets.

2. Is the process tuned for each perovskite composition? For example, is the antisolvent quenching time optimized? The ink formation influences the crystallization process, which is likely to impact wrinkle formation and the overall film quality. Is there a relationship between the pH of the precursor and the film crystallization?

3. Page 13: Authors discuss the “illuminated” samples to investigate the halide segregation in the films. The stated that the samples were continuously illuminated for 5 min. The details of the illumination condition comes in page 15. Please consider to move the details in page 13 where the first time the experiment mentioned. This would be easier for the readers.

4. In Figure 5, for the PL redshift images, please consider adjusting the resolution of the scale bar. The 3 different color scale represents a nearly 40 nm shift, which corresponds to a significant change in bandgap. A higher resolution PL peak wavelength image after 5 min of illumination would more effectively highlight the differences although I believe the histograms already show the differences before and after continuous illumination.

5. How long did the PL mapping measurement take? The experimental section mentions an exposure time of 1 minute 15 seconds per frame. Did the authors test whether this exposure time initiates halide segregation in samples with high Br content?

6. I understand why authors did not discuss the device parameters in the main text. However, the quantification of these losses is missing. They demonstrated that Urbach energy increases in samples with greater heterogeneity. Including a Urbach energy vs. Woc graph (Fig 3 in the following paper: <https://pubs.acs.org/doi/full/10.1021/acs.jpcclett.2c01812>) for these samples would better correlate the defects with device performance. Additionally, in Figure 6c, higher Br content increases the Urbach energy rather than decreasing it. Lower Urbach energy indicates better electronic quality, so please correct the figure accordingly.

Reviewer #3

(Remarks to the Author)

Comments

The manuscript on the whole looks well-organized and thorough. The results showcased in the manuscript are mostly straightforward to understand and complement each other along with validating the conclusions. The manuscript needs some edits and refinement to better present the results.

Introduction

Page 3, Line 2 Introducing mixed-halide wide-bandgap perovskites as only APbX₃ might not be the right way as there are wide bandgap perovskites reported with no Pb and only Sn in them - <https://pubs.acs.org/doi/10.1021/acsenergylett.4c00796>, So a generic representation as ABX₃ might be better suited. The introduction section is well ordered explaining the reason for looking into wide bandgap perovskites and the reasons for having compositional and morphological heterogeneity. It then explains why a correlation between compositional and morphological heterogeneity in wide band-gap compositions is needed and how this work achieves that.

Results

Figure 1h – can't say if the color in the figure matches the color scale of the image or not. The color in the figure looks a bit greenish.

Three-dimensional AFM profiles for the compositions look great but these profiles can be included for all the compositions for better visualization to the audience and better consistency in the supplementary information.

Page 6, Line 25 – Instead of mentioning the feature size of the smooth films to be around 100 nm it would be better if mentioned in μm , this would distinguish the feature sizes from smooth to rough films along with easy comparison

Page 6, Line 26 – It is confusing if different names are used for the rough films, such as heterogeneous films, a common representation can be used.

Page 6, Line 28 – A description can be added for the increase in average roughness with the increase in MA and Br content as seen in Fig.1 and Supplementary Fig.4

Why was the compositional study not performed with MA content of 0% and 100%? Even though they do not fall in the desired bandgap of the perovskite top won't the insights be helpful to check if the trend of large feature sizes continues with more MA content?

Page 8, Line 14 – Roughness information for 50% Br compositions i.e. 0.25/0.50, 0.50/0.50, 0.75/0.50 can be added to supplementary as it is not present in either Fig.1 or the supplementary figures.

Page 8, Line 18 – A reference to Fig.2c, 2e, 2g is missing in the main text

Supplementary Fig.6c is labeled wrong in the caption as 0.25/0.40, it should be 0.25/0.50

Figure 3 – check the spin coating timing in methods

Page 9, Line 11 – Is there a reason why the time stamp between 25s and 31s is different for Fig.3 (29s) and Supplementary Fig.7 (27s)?

Supplementary Fig.7 – 0.75/0.50 does not show similar features (as in the yellow shading) as Fig.3 for the antisolvent drop at 25s, This figure needs to be checked if the images are misplaced or if there is an actual difference.

Also, there is a discrepancy in the labeling of Supplementary Fig.7 and its captions.

Fig.3 – It is evident from the figure but it is better to mention that Fig.3a, c, e, g are of 0.25/0.60, and Fig.3b, d, f, g are of 0.75/0.60 in the caption.

Explanation and reference for Fig. 3i and 3j are missing in the main text.

Page 11, Line 3 – The sentence on the elemental map of Pb is confusing, can be simplified

Page 11, Line 13 – It would be better to mention it as homogeneous distribution of Pb, Br, and I instead of homogeneous distribution of ions

Fig.4 – Line cuts can be elaborated in a better way in the main text, what are the dotted lines in those figures?

Supplementary figure 12 – homogeneous distribution of ions? 12a – has more iodine rich regions, 12b more equal variation, 12c much smoother variation

Supplementary figure 13 – Instead of homogeneous and heterogeneous distribution can this be called clustered distribution vs unclustered distribution?

Page 15, Line 10 – Hyperspectral PL study on composition 0.75/0.60 can be included to make a stronger claim on the statement about an increase in PL heterogeneity with an increase in bromide content. The same study can be included for composition 0.25/0.60 to rule out the possibility of PL heterogeneity increase with Br at lower MA content.

Fig.6 – In the figure caption the continuous illumination is mentioned as 10 minutes but in the main text it is mentioned as 5 minutes – check this discrepancy

The claims over increased photostability based on the inclusion of MA without any device or accelerated testing under light + heat is a bit concerning, especially due to the well-characterized instability challenges of MA inclusion. How can the authors be sure that the improvements in morphology from crystallization control with larger MA fractions will not lead to degradation in devices?

There is a discussion of relevance to tandems and multi-junctions in the work—such as claiming the wrinkled surface is not amenable to multijunction processing—but there are reports of high-efficiency tandems on textured Si with a similar profile to the wrinkled films (<https://www.sciencedirect.com/science/article/pii/S2542435120300350>). Can the authors comment on why they expect this to be different than in the case of Si?

The results section has the required details, which are well-elaborated. The hypothesis observed is well-explained and well-validated by relevant tests. However, the results section still needs some refinement to put forth the best representation.

Summary

This section is well-explained.

Materials and Methods

Well-elaborated.

Version 1:

Reviewer comments:

Reviewer #1

(Remarks to the Author)

I thank the reviewers for addressing my comments. One lingering question remains, though. The authors' explanation of wrinkle formation due to the instability of the elastic/viscoelastic bilayer is reasonable. However, what remains unexplained is the relative richness of the peaks/ridges in iodide and the troughs/valleys in bromide.

Reviewer #2

(Remarks to the Author)

The authors have responded to the reviewers' comments and revised the manuscript accordingly. I have no additional remarks.

Reviewer #3

(Remarks to the Author)

The comments and the suggestions were addressed adequately.

RESPONSE TO REVIEWERS

We thank the reviewers for their careful consideration of our work and their constructive feedback. We have addressed the comments and questions in our responses below. The comments/questions from reviewers are marked in **BLUE**, the authors' responses are marked in **BLACK** and any references to the Main Text or Supplementary Information are marked in **RED**. Changes in the Main Text and Supplementary Information files are highlighted in **yellow**. We note that since several Supplementary Figs. have been changed in sequence, they have also been highlighted.

Reviewer #1 (Remarks to the Author):

“Local halide heterogeneity drives surface wrinkling in mixed-halide wide-bandgap perovskites” by Kunal Datta et al. discusses that wrinkling in $(\text{FA}_{1-x}\text{MA}_x)\text{Pb}(\text{I}_{1-y}\text{Br}_y)_3$ perovskite is compositionally driven, requiring high concentrations of MA, and found that the peaks of these wrinkles are iodide-rich while the troughs are bromide-rich. This conclusion is interesting and well-supported by the experimental results. There are a few lingering questions remaining that should be addressed before this manuscript is published, though.

We thank the reviewer for their positive evaluation.

1. One general comment given how many compositions the authors tested was that it was hard to keep straight which compositions were wrinkled and which weren't as I was reading. Perhaps there could be some sort of additional designator in addition to the x/y to help out people like me from having to refer back to figure 1 all the time?

Thank you for raising this very important issue. We have considered the feedback and have changed the way we designate the different compositions in the body of the text and in figure captions both in the main text file and the Supplementary Information. We now indicate smooth samples as $x|y$ and wrinkled samples as $x:y$ and an intermediate roughness as $x|y$. For example, the $0.25|0.40$ composition yields a smooth film, the $0.75:0.60$ composition yields a wrinkled film and the $0.25|0.60$ film has an intermediate behavior. We hope this change helps readability. We have also added a piece of text in the main body to explain this nomenclature (see below).

Page 6: “Hereafter, we refer to these compositions as $\{x|y\}$ for smooth films, $\{x:y\}$ for rough films, and $\{x|y\}$ for films with intermediate morphology. For example, $\{x/y\} = 0.25|0.40$ forms a smooth film, $\{x/y\} = 0.75:0.60$ forms a rough film, and $\{x/y\} = 0.25|0.60$ has an intermediate behavior.”

2. It is difficult to tell if the PbI_2 peak is unchanged in the GIWAXS as stated by the authors. There appears to be an orientational change depending on the composition.

Thank you for this interesting question. For the PbI_2 peak, we can see from the 2D GIWAXS images that the position ($q \sim 0.9 \text{ \AA}^{-1}$) is unchanged irrespective of film composition. We have also analyzed the orientation through azimuthal integration of the diffraction feature. As can be seen (figure below), the preferential orientation undergoes some changes in different compositions. In particular, the MA content seems to have a noticeable impact on the orientation wherein increasing the MA content increases preferential orientation. In contrast, however, the Br content does not

have a strong influence on the orientation even though the Br content has a stronger impact on the wrinkling behavior than the MA content. Therefore, although there are some changes in the orientation, we conclude that they may not be completely related to the wrinkling behavior we report in this manuscript. The main text has been edited to note the change in orientation.

Page 7: “The peak position corresponding to unreacted PbI_2 ($q \approx 0.9 \text{ \AA}^{-1}$) remains unchanged across compositions, albeit with an increase in preferential orientation with increasing MA content (Supplementary Fig. 9).”

Supplementary Fig. 9 | Orientation of PbI_2 from GIWAXS patterns of $(\text{FA}_{1-x}\text{MA}_x)\text{Pb}(\text{I}_{1-y}\text{Br}_y)_3$ perovskites. Azimuthal intensity profiles of the Debye-Scherrer ring associated to PbI_2 ($q = 0.9 \text{ \AA}^{-1}$) as a function of χ angle from GIWAXS for perovskite compositions with Br content **a** 40%, **b** 50%, and **c** 60%. The MA content is varied between 25%, 50%, and 75%.

3. Also in the GIWAXS, the authors refer to “Fig. 2f, 2h, 2i, 2j and 2l” when discussing intense spots on the ring at $q = 1.0 \text{ \AA}^{-1}$, but it does not seem like all of these figures align with what is written.

Thank you for pointing it out. It was a typo and has been corrected to exclude 2j and 2l. Please see below for the modified text.

Page 9: “the appearance of intense spots on the ring at $q \approx 1.0 \text{ \AA}^{-1}$ (Fig. 2f, 2h, and 2i), especially in the out-of-plane”

4. The correlation between the PL and TOF-SIMS is not presented in a particularly conclusive way. I can maybe see what the authors are stating if I squint, but I could just be seeing what the authors want me to see. Is there a better way to present this information?

Thank you for this question. We have processed the TOF-SIMS data in Supplementary Fig. 21 (previously Supplementary Fig. 17) further using Gaussian smoothening that has reduced the amount of detail and helped make the data easier to understand (see new figure below). We have also marked two other locations in the frame where we can find an association between the PL emission and the halide composition. We hope the correlation is easier to perceive now.

Supplementary Fig. 21 | PL and TOF-SIMS map of pristine $(\text{FA}_{1-x}\text{MA}_x)\text{Pb}(\text{I}_{1-y}\text{Br}_y)_3$ perovskite film. a PL intensity map (750 nm) for perovskite composition $\{x/y\} = 0.75:0.50$. Dashed square represents sample area was used for TOF-SIMS measurement. **b** Corresponding TOF-SIMS map of I^-/Br^- ratio.

5. A key question remaining in this work is are these wrinkles due to film compression and subsequent delamination? If so, what mechanism would be driving volumetric expansion of the film as it is drying? Is it possible that the wrinkles occur where the perovskite is iodide-rich due to the larger size of iodide forcing the perovskite to distort out-of-plane to relax stresses?

Thank you for raising this question. Isotropic wavy pattern wrinkles similar to those described in the manuscript have been observed in several thin films, e.g., sol-gel derived zinc oxide (*Phys. Rev. E* **2005**, *71*, 011604 and *Colloids Surf., A* **2023**, *658*, 130628) and alumina (*Appl. Phys. Lett.* **2016**, *108*, 151601), in $\text{YBa}_2\text{Cu}_3\text{O}_{7-x}$ (*Appl. Surf. Sci.* **2015**, *355*, 736–742), and perovskites (*ACS Energy Lett.* **2018**, *3*, 1225–1232; *Nat. Commun.* **2021**, *12*, 1554). Their formation has been described theoretically (*J. Appl. Mech.* **2004**, *72*, 955-961) in terms of a bilayer system consisting of an elastic layer, on top of a viscoelastic layer, on top of a rigid substrate. Our findings agree with the proposed mechanism and we explain the wrinkle formation as follows. The perovskite precursor solution is initially present as a homogeneous elastic layer. Immediately after the antisolvent is cast, a Br-rich perovskite phase in the presence of higher MA concentrations, forming a skin layer with a higher elastic modulus than the wet film below. As the remaining more I-rich film below dries and shrinks it exerts a compressive stress on the top layer. The wrinkling is then a consequence of a spinodal-like decomposition which creates a pattern of ridges and valleys (*Phys. Rev. Lett.* **2003**, *91*, 154502). The valleys are mainly rich in bromide and the ridges are rich in iodide. This entire process sets in within approx. the first 5 s of film crystallization (immediately after antisolvent cast) while the film is still wet. While subsequent annealing promotes crystallization, the orientation and wrinkling has largely occurred already (Supplementary Fig. 13).

Page 21-22: “Isotropic wavy pattern wrinkles similar to those described in the manuscript have been observed in several thin films, e.g., sol-gel derived zinc oxide and alumina, in $\text{YBa}_2\text{Cu}_3\text{O}_{7-x}$, and perovskites^{16,18,59-61}. Their formation has been described theoretically in terms of a bilayer system consisting of an elastic layer, on top of a viscoelastic layer, on top of a rigid substrate⁶².

Our findings agree with the proposed mechanism and we rationalize the wrinkle formation as follows. The perovskite precursor solution is initially present as a homogenous elastic layer. Immediately after the antisolvent is cast a Br-rich perovskite phase crystallizes in the presence of higher MA concentrations, forming a skin layer with a higher elastic modulus than the wet film below. As the remaining more I-rich film below dries and shrinks it exerts a compressive stress on the top layer. The wrinkling is then a consequence of a spinodal-like decomposition which creates a pattern of ridges and valleys⁶³. The valleys are mainly rich in bromide and the ridges are rich in iodide. This entire process sets in within approx. the first 5 s of film crystallization (immediately after antisolvent cast) while the film is still wet. While subsequent annealing promotes crystallization, the orientation and wrinkling has largely occurred already (Supplementary Fig. 13).”

Reviewer #2 (Remarks to the Author):

Datta et al. reported an in-depth analysis of compositional heterogeneity in wide bandgap perovskites and its impact on their structural and optoelectronic properties. Authors analyzed different set of samples (varying the A-site cation and halides, respectively) to investigate the wrinkle formation on the surface of the films. The local inhomogeneities (morphological and compositional) were identified as key contributors to film degradation under continuous light exposure. The findings of the significant and timely for research on mixed cation mixed halide perovskites.

We thank the reviewer for their positive evaluation and for recognizing the relevance of this work in the current context of the field.

1. Page 4: Authors used different FA and MA ratios for A-site cation, and different bromide and iodide ratios for X-site halide effects. In sample labeling, denoting as x/y was hard to follow in the main text. Directly MA/Br percentages would be easier to follow as in figure insets.

Thank you for pointing this out. We have added a small piece of text (see below) at the beginning of the results section to explain what x and y represent. Additionally, as also suggested by Reviewer #1, we have changed the way we designate compositions. We now indicate smooth samples as $x|y$ and wrinkled samples as $x:y$ and an intermediate roughness as $x|y$. For example, the 0.25|0.40 composition yields a smooth film, the 0.75:0.60 composition yields a wrinkled film and the 0.25|0.60 film has an intermediate behavior. This change has been made throughout the main text and supplementary information text and figure captions and we hope it helps readability. We have also added a piece of text in the main body to explain this nomenclature.

Page 4: “where x refers to MA content and y refers to Br content”

Page 6: “Hereafter, we refer to these compositions as $\{x|y\}$ for smooth films, $\{x:y\}$ for rough films, and $\{x|y\}$ for films with intermediate morphology. For example, $\{x|y\} = 0.25|0.40$ forms a smooth film, $\{x/y\} = 0.75:0.60$ forms a rough film, and $\{x/y\} = 0.25|0.60$ has an intermediate behavior.”

2. Is the process tuned for each perovskite composition? For example, is the antisolvent quenching

time optimized? The ink formation influences the crystallization process, which is likely to impact wrinkle formation and the overall film quality. Is there a relationship between the pH of the precursor and the film crystallization?

Processing variables are not tuned for each film since we wanted to study the direct impact of the perovskite composition. We have now clearly stated this in the Methods section (see text below). During this study, however, we studied the impact of some variables on wrinkling behavior but found no significant differences. For example, we have studied the impact of surface hydrophobicity by depositing the perovskite on a hydrophobic PTAA layer (*Nat. Commun.* **2015**, 6, 7747). However, wrinkling behavior was unchanged (see figure below). Furthermore, we have studied the impact of annealing conditions ranging from no annealing to 5 min annealing. In all cases, the wrinkling behavior was unchanged (see figure below), further supporting our hypothesis that wrinkle formation starts to occur during spin-coating and is largely unaffected by annealing conditions.

The pH of the precursor solution has previously been studied in relation to effect of the acidic hydrolysis and thermal decomposition of DMF on the crystallization and optoelectronic properties of perovskite films (*Joule* **2017**, 1, 2, 328 – 343). Formamidinium is only slightly less acidic than (pKa (FA⁺) ≈ 11.5) than methylammonium (pKa (MA⁺) ≈ 10.6) (*Nat. Energy* **2023**, 8, 1229 – 1239, *J. Phys. Chem. C* **2021**, 125, 21851 – 21861). Hence, the acidities of the precursor solutions are very similar and an effect on wrinkling is not expected.

Impact of surface hydrophobicity on wrinkling | Surface SEM images of perovskite thin films with 50% MA content and 60% Br content deposited on **a** glass and **b** PTAA.

Impact of annealing time on wrinkling | Surface SEM images of perovskite thin films with 50% MA content and 60% Br content annealed for **a** 0 s, **b** 10 s, **c** 1 min and **d** 5 min.

Page 23: “It must be noted that processing methods are unchanged across all compositions.”

3. Page 13: Authors discuss the “illuminated” samples to investigate the halide segregation in the films. The stated that the samples were continuously illuminated for 5 min. The details of the illumination condition comes in page 15. Please consider to move the details in page 13 where the first time the experiment mentioned. This would be easier for the readers.

We thank the reviewer for this comment. We have added the experimental conditions (wavelength, intensity) for illumination to an earlier part of the text (see below).

Page 14: “We then continuously illuminated the film for another 5 min. (450 nm, 130 mW cm⁻²), and measured again with a 1 min. acquisition time (referred to as “illuminated”).”

4. In Figure 5, for the PL redshift images, please consider adjusting the resolution of the scale bar. The 3 different color scale represents a nearly 40 nm shift, which corresponds to a significant change in bandgap. A higher resolution PL peak wavelength image after 5 min of illumination would more effectively highlight the differences although I believe the histograms already show the differences before and after continuous illumination.

Thank you for this suggestion. We have changed the scale bars in Figs. 5b, 5f and 5j and used a more appropriate color scale to better highlight the local differences in PL redshift. Please see figure below.

Fig. 5 | Heterogeneity in photoluminescence emission. Hyperspectral luminescence of perovskite thin films with compositions. **a – d** $\{x/y\} = 0.25|0.40$. **e – h** $\{x/y\} = 0.75:0.40$. **i – l** $\{x/y\} = 0.75:0.50$. Here, 2D emission maps in panels **a**, **e**, and **i** represent the wavelength at emission maximum for pristine films. 2D maps in panels **b**, **f**, and **j** show the wavelength change ($\Delta\lambda$) upon continuous illumination (450 nm, 5 min.). Spectra in panels **c**, **g**, and **k** are averaged over the scanned area of pristine (red line) and illuminated (blue shaded) films. Panels **d**, **h**, and **l** show histogram of maximum emission wavelengths in pristine (red) and illuminated (blue) thin films. The maps show that emission heterogeneity increases with increasing MA and Br content and that

in rough films, regions of low-energy emission undergo a smaller redshift after continuous illumination. Note the different color scales in panels **b**, **f** and **j**.

5. How long did the PL mapping measurement take? The experimental section mentions an exposure time of 1 minute 15 seconds per frame. Did the authors test whether this exposure time initiates halide segregation in samples with high Br content?

One limitation of the PL acquisition technique was that the minimum acquisition time (1 min) restricts our ability to capture events occurring on a shorter timescale. Therefore, in order to observe larger changes, we conducted the 5 min light exposure stress experiment and observe the progression of halide segregation over a longer period. We cannot exclude that in some cases, for compositions with high Br content, some ion migration may have happened within the 1 min acquisition time.

6. I understand why authors did not discuss the device parameters in the main text. However, the quantification of these losses is missing. They demonstrated that Urbach energy increases in samples with greater heterogeneity. Including a Urbach energy vs. V_{oc} graph (Fig 3 in the following paper: <https://pubs.acs.org/doi/full/10.1021/acs.jpcclett.2c01812>) for these samples would better correlate the defects with device performance. Additionally, in Figure 6c, higher Br content increases the Urbach energy rather than decreasing it. Lower Urbach energy indicates better electronic quality, so please correct the figure accordingly.

Thank you for this suggestion. We have considered this representation (Urbach energy vs V_{oc} loss) as well for this work. However, a key issue in calculating the Urbach energy here is that halide heterogeneity in wrinkled perovskites introduces additional features in the band tail (Fig. 6c). This means that the commonly used method of fitting the exponential tail (*Nat. Commun.* **2017**, 8, 590, *Adv. Energy Mater.* **2020**, 10, 1902573) is no longer applicable due to the presence of several absorbing species and instead of one Urbach energy, the film has a distribution of Urbach energies (similar to Fig. 2b in *Nat. Nanotechnol.* **2022**, 17, 190–196) which cannot be estimated at the cell level using the photocurrent spectroscopy method we are using. As a result, while our current methods are designed to provide an understanding of average band-edge disorder, microscopy techniques would be needed to precisely associate local changes of Urbach energy to local V_{oc} losses. We have mentioned the relevance of such methods in the main text (see text below). Also, thank you for pointing out the typo in Fig. 6c, we have corrected it.

Page 18: “Increasing variations in local bandgap-edge have also been related to non-radiative recombination driven by sub-bandgap defects³².”

Reviewer #3 (Remarks to the Author):

Comments:

The manuscript on the whole looks well-organized and thorough. The results showcased in the manuscript are mostly straightforward to understand and complement each other along with validating the conclusions. The manuscript needs some edits and refinement to better present the results.

We thank the reviewer for their positive feedback on our work.

Introduction

Page 3, Line 2 Introducing mixed-halide wide-bandgap perovskites as only APbX₃ might not be the right way as there are wide bandgap perovskites reported with no Pb and only Sn in them - <https://pubs.acs.org/doi/10.1021/acsenergylett.4c00796>, So a generic representation as ABX₃ might be better suited.

Thank you for pointing that out. We have changed this in the Introduction (see below).

Page 3: “Mixed-halide wide-bandgap ABX₃ (A is a monovalent cation, B is lead or tin, and X is a halide ion (iodide or bromide)) perovskite semiconductors are promising candidates for use in monolithic multijunction photovoltaic devices where the use of complementary absorber layers enables an increase in photovoltaic performance¹.”

The introduction section is well ordered explaining the reason for looking into wide bandgap perovskites and the reasons for having compositional and morphological heterogeneity. It then explains why a correlation between compositional and morphological heterogeneity in wide band-gap compositions is needed and how this work achieves that.

We thank the reviewer for their positive evaluation of the Introduction section and how it lays out the background for this research.

Results

Figure 1h – can't say if the color in the figure matches the color scale of the image or not. The color in the figure looks a bit greenish.

Thank you for pointing that out. We have corrected the 3D AFM figure panels (see figure below). We have also recalculated the R_p values based on the 2D images (Supplementary Fig. 5) and the new values are reflected in panels j, k, l, m, n and o of Fig. 1. The trends are the same as before.

Three-dimensional AFM profiles for the compositions look great but these profiles can be included for all the compositions for better visualization to the audience and better consistency in the supplementary information.

Thank you for the suggestion. We have added the 3D AFM maps (see below) for all nine compositions in the Supplementary Information (Supplementary Fig. 6).

Supplementary Fig. 6 | Three-dimensional atomic force microscopy height profiles of $(\text{FA}_{1-x}\text{MA}_x)\text{Pb}(\text{I}_{1-y}\text{Br}_y)_3$ perovskite thin films. a $\{x/y\} = 0.25|0.40$. b $\{x/y\} = 0.50|0.40$. c $\{x/y\} = 0.75|0.40$. d $\{x/y\} = 0.25|0.50$. e $\{x/y\} = 0.50|0.50$. f $\{x/y\} = 0.75|0.50$. g $\{x/y\} = 0.25|0.60$. h $\{x/y\} = 0.50|0.60$. i $\{x/y\} = 0.75|0.60$.

Page 6, Line 25 – Instead of mentioning the feature size of the smooth films to be around 100 nm it would be better if mentioned in μm , this would distinguish the feature sizes from smooth to rough films along with easy comparison

We have changed the units from nm to μm when discussing feature sizes observed in AFM measurements (see below).

Page 7: “The feature sizes in the smooth films ($\{x/y\} = 0.25|0.40$, $0.50|0.40$, and $0.25|0.60$) are on the order of $0.10 \mu\text{m}$ whereas compositions that yield rough films ($\{x/y\} = 0.75|0.40$, $0.50|0.60$, and $0.75|0.60$) exhibit feature sizes as large as $1.5 - 2.0 \mu\text{m}$.”

Page 6, Line 26 – It is confusing if different names are used for the rough films, such as heterogeneous films, a common representation can be used.

Thank you for this suggestion. We’ve made changes to the text to distinguish between morphological behavior and compositional/photoluminescence properties. We have now used the words “smooth” and “rough” when discussing morphology and we use “homogeneous” and “heterogeneous” when describing other forms of disorder such as for the XRF, PL and TOF-SIMS data.

Page 6, Line 28 – A description can be added for the increase in average roughness with the increase in MA and Br content as seen in Fig.1 and Supplementary Fig.4 Why was the compositional study not performed with MA content of 0% and 100%? Even though they do not fall in the desired bandgap of the perovskite top won't the insights be helpful to check if the trend of large feature sizes continues with more MA content?

To address the first part of this comment, we have edited the main text (page 6) to describe the increase in roughness parameters R_p and R_q .

Page 7: “This results in an increase in the average maximum profile height (R_p) and root mean square average roughness (R_q) as a function of increasing MA and Br contents (Supplementary Fig. 7).”

We have also studied a wider range of MA-contents for Br-contents 40% and 60%. SEM images (see below) show that the trends are similar; lower MA content films are smooth while high MA content films are rough. The figure has been added to the Supplementary Figures and the main text has been edited to refer to it (see text below).

Page 6: “We note that the behavior is also consistent at lower (0%) and higher (100%) MA contents (Supplementary Fig. 3) and that the appearance of morphological disorder does not significantly affect SEM features at 1 – 2 μm size or create voids in the film surface (Supplementary Fig. 4).”

Supplementary Fig. 3 | Surface SEM images of $(\text{FA}_{1-x}\text{MA}_x)\text{Pb}(\text{I}_{1-y}\text{Br}_y)_3$ perovskite thin films for different compositions. a $\{x/y\} = 0|0.40$. b. $\{x/y\} = 1.0|0.40$. c. $\{x/y\} = 0|0.60$. d. $\{x/y\} = 1.0|0.60$.

Page 8, Line 14 – Roughness information for 50% Br compositions i.e. 0.25/0.50, 0.50/0.50, 0.75/0.50 can be added to supplementary as it is not present in either Fig.1 or the supplementary figures.

We have added SEM and AFM data for 50% Br series to the Supplementary Information (Supplementary Figs. 2, 5 and 6) along with values for R_p and R_q (Supplementary Fig. 7) (see figures below).

Supplementary Fig. 2 | Surface SEM images of $(FA_{1-x}MA_x)Pb(I_{1-y}Br_y)_3$ perovskite films with 50% Br content and different MA contents. a $x = 0.25$. b. $x = 0.50$. c. $x = 0.75$.

Supplementary Fig. 5 | Atomic force microscopy height profiles of $(FA_{1-x}MA_x)Pb(I_{1-y}Br_y)_3$ perovskite thin films. a $\{x/y\} = 0.25|0.40$. b $\{x/y\} = 0.50|0.40$. c $\{x/y\} = 0.75|0.40$. d $\{x/y\} = 0.25|0.50$. e $\{x/y\} = 0.50|0.50$. f $\{x/y\} = 0.75|0.50$. g $\{x/y\} = 0.25|0.60$. h $\{x/y\} = 0.50|0.60$. i $\{x/y\} = 0.75|0.60$. Scale bars are 10 μm . Height range is from 0 – 2.0 μm .

Supplementary Fig. 6 | Three-dimensional atomic force microscopy height profiles of $(\text{FA}_{1-x}\text{MA}_x)\text{Pb}(\text{I}_{1-y}\text{Br}_y)_3$ perovskite thin films. a $\{x/y\} = 0.25|0.40$. b $\{x/y\} = 0.50|0.40$. c $\{x/y\} = 0.75|0.40$. d $\{x/y\} = 0.25|0.50$. e $\{x/y\} = 0.50|0.50$. f $\{x/y\} = 0.75|0.50$. g $\{x/y\} = 0.25|0.60$. h $\{x/y\} = 0.50|0.60$. i $\{x/y\} = 0.75|0.60$.

Supplementary Fig. 7 | Film roughness as a function of $(\text{FA}_{1-x}\text{MA}_x)\text{Pb}(\text{I}_{1-y}\text{Br}_y)_3$ perovskite composition. a Average maximum peak profile height. b Root mean square average roughness.

Page 8, Line 18 – A reference to Fig.2c, 2e, 2g is missing in the main text
 Supplementary Fig.6c is labeled wrong in the caption as 0.25/0.40, it should be 0.25/0.50

Thank you for pointing that out. We have made the corrections (see below).

Page 9: “smooth films ($\{x/y\} = 0.25|0.40, 0.50|0.40, 0.25|0.50, 0.50|0.50, \text{ and } 0.25|0.60$) show GIWAXS patterns (Fig. 2a, 2b, 2d, 2e, and 2g)”

Page 9: “roughness $> 1 \mu\text{m}$ with composition ($\{x/y\} = 0.75\text{:}0.40, 0.75\text{:}0.50, 0.50\text{:}0.60, \text{ and } 0.75\text{:}0.60$) show GIWAXS patterns”

Figure 3 – check the spin coating timing in methods

We have confirmed the spin-coating time. To clarify, the data presented in Figure 3 shows a small section (between 21 s and 31 s) of the overall 40 s spin-coating period. The structural evolution during the entire process can be seen in Supplementary Fig. 13.

Page 9, Line 11 – Is there a reason why the time stamp between 25s and 31s is different for Fig.3 (29s) and Supplementary Fig.7 (27s)?

Thank you for pointing this out. We have now made the two figures uniform, including the frame at 29 s in the Supplementary Figure (see below). The structural behavior is similar to what we had previously reported at 27 s.

Supplementary Fig. 11 | In situ GIWAXS patterns acquired at different times during spin-coating of $(\text{FA}_{1-x}\text{MA}_x)\text{Pb}(\text{I}_{1-y}\text{Br}_y)_3$ perovskite films. a, c, e and g $\{x/y\} = 0.25|0.50$. b, d, f and h and $0.75:0.50$. The anti-solvent is applied after 25 s (panels c and d).

Supplementary Fig.7 – $0.75/0.50$ does not show similar features (as in the yellow shading) as Fig.3 for the antisolvent drop at 25s, This figure needs to be checked if the images are misplaced or if there is an actual difference.

The yellow shaded region in Supplementary Fig. 11 is related to scattering observed when the antisolvent is cast onto the film but disappears immediately after casting. Similar increases in intensity in the $q \approx 1 - 1.2 \text{ \AA}^{-1}$ range has previously been reported in other studies using in-situ GIWAXS measurements (for example Fig. 3 in *Adv Energy Mater.* **2017**, 7, 1602600, Figs. 2b and 2f in *Adv. Mater.* **2019**, 31, 1901284). A likely explanation for its absence in the $\{x/y\} = 0.75/0.50$ measurement is the ms-scale mismatch in the antisolvent drip timing and the frame acquisition which likely causes this feature to not appear prominently in the 25 s frame.

Also, there is a discrepancy in the labeling of Supplementary Fig.7 and its captions.

There was an error in the panel labels in the caption, it has been rectified (see below). Thank you for pointing it out.

Page S8: “**Supplementary Fig. 11 | In situ GIWAXS patterns acquired at different times during spin-coating of $(\text{FA}_{1-x}\text{MA}_x)\text{Pb}(\text{I}_{1-y}\text{Br}_y)_3$ perovskite films. a, c, e, and g $\{x/y\} = 0.25|0.50$. b, d, f, and h and $0.75:0.50$. The anti-solvent is applied after 25 s (panels c and d).**”

Fig.3 – It is evident from the figure but it is better to mention that Fig.3a, c, e, g are of $0.25/0.60$, and Fig.3b, d, f, g are of $0.75/0.60$ in the caption.

Thank you for the suggestion, we have made the suggested change in the figure caption.

Page 11: “**Fig. 3 | Crystallization dynamics of mixed-halide perovskite thin films. In situ GIWAXS patterns of perovskites with compositions $\{x/y\} = 0.25|0.60$ and $\{x/y\} = 0.75:0.60$ during spin-coating. Panels mark time stamps during the spin coating process. a, b 21 s. c, d 25 s. e, f 29 s. g, h 31 s. The frame at 25 s represents the casting of the antisolvent onto the substrate. Azimuthal intensity profiles of the main Debye-Scherrer ring (100) as a function of χ angle from GIWAXS for different perovskite compositions acquired in the 20 – 40 s period of spin-coating. i $\{x/y\} = 0.25|0.60$. j $\{x/y\} = 0.75:0.60$. Panels a, c, e, g, and i refer to $\{x/y\} = 0.25|0.60$ and panels b, d, f, h, and j refer to $\{x/y\} = 0.75:0.60$. The data in panels i and j have been vertically offset for clarity.**”

Explanation and reference for Fig. 3i and 3j are missing in the main text.

Thank you for pointing that out, we have referred to it in the main text. The explanation was already written (referring to Supplementary Fig. 9) in the original version of the text. Now it also refers to Fig. 3i and 3j (see below).

Page 10: “we propose that in an MA-rich environment, heterogeneous crystal nucleation leads to the formation of oriented bromide-rich perovskites immediately after antisolvent casting (Fig. 3i, 3j and Supplementary Fig. 13) followed by the incorporation of iodide-containing phases¹⁸”

Page 11, Line 3 – The sentence on the elemental map of Pb is confusing, can be simplified

Thank you for the suggestion. We have added a sentence to explain the relationship between Pb content and film thickness (see below).

Page 11-12: “As a result, regions with higher Pb content refer to the peak-like regions of wrinkles whereas lower Pb content corresponds to valley-like regions.”

Page 11, Line 13 – It would be better to mention it as homogeneous distribution of Pb, Br, and I instead of homogeneous distribution of ions

Thank you for the suggestion, the text has been changed (see below) to be more specific to the elements being discussed.

Page 12: “Similar homogeneous distribution of iodide-to-bromide ratio is observed for other smooth films”

Fig.4 – Line cuts can be elaborated in a better way in the main text, what are the dotted lines in those figures?

Thank you for the suggestion. We have marked the lines as solid and dashed for iodide-to-bromide ratio and Pb-content respectively in the figure caption (see below). We have also emphasized in the main text that the correlation being drawn is between the local increase in layer thickness (Pb content) and higher iodide concentration.

Page 13: “**Fig. 4 | Compositional heterogeneity in mixed-halide perovskite thin films from nano-XRF.** **a** Normalized elemental maps of Pb, I, and Br of perovskite thin film with $\{x/y\} = 0.25/0.40$. **b** Map of iodide-to-bromide ratio in shaded region of panel **a**. Sub-panels A, B, C, and D represent the normalized Pb elemental map in regions highlighted (dashed squares) in the iodide-to-bromide map. **c** Line cuts of iodide-to-bromide ratios (solid) overlapped with local Pb content line cuts (dashed) marked with (1) and (2) in panel **b**. **d** Normalized elemental maps of Pb, I, and Br of perovskite thin film with $\{x/y\} = 0.50/0.60$. **e** Map of iodide-to-bromide ratio in shaded region of panel **d**. Sub-panels E, F, G, and H represent the normalized Pb elemental map in regions highlighted in the iodide-to-bromide map. **f, g** Line cuts of iodide-to-bromide ratios (solid) overlapped with local Pb content line cuts (dashed) at points marked with (1), (2), (3), and (4) in panel **e**. Maps show that smooth films yield homogeneous halide distribution across the film thickness and wrinkled films have iodide-rich domains concentrated at peak-like regions. All scale bars are 10 μm .”

Page 12: “Line cuts at four distinct locations of the film and two-dimensional maps (Fig. 4f, 4g, and Supplementary Fig. 18) further confirm the positive correlation between the local increase in layer thickness (Pb content) and higher iodide concentration.”

Supplementary figure 12 – homogeneous distribution of ions? 12a – has more iodine rich regions, 12b more equal variation, 12c much smoother variation

Thank you for this comment. We would like to clarify that the three compositions in Supplementary Fig. 16 $\{x/y\} = 0.50/0.40, 0.50/0.50$ and $0.25/0.60$ have different halide contents as a result of which the overall iodide/bromide content is different in each case. There is also a stochastic distribution of halides which may result in nanometer-scale domains rich in either iodide/bromide as has also been observed by others (*J. Am. Chem. Soc.* **2016**, 138, 15821 – 15824 and *ACS Energy Lett.* **2022**, 7, 471 – 480). What we would like to emphasize in this work though that in these compositions, the large micrometer-scale domains related to wrinkling are absent. We have clarified this in the main text now (see text below) and thank you for raising this issue.

Page12: “We note here that the stochastic distribution of ions causes local nanometer-scale domains to develop that are rich in iodide or bromide ions^{26,28}.”

Supplementary figure 13 – Instead of homogeneous and heterogeneous distribution can this be called clustered distribution vs unclustered distribution?

Thank you for this suggestion. Indeed the nomenclature referring to this behavior can use several combination of words such as homogeneous/heterogeneous, homogeneous/inhomogeneous, mixed/unmixed, segregated/non-segregated, aggregated/disaggregated and clustered/unclustered. And although we understand the motivation to use “clustered/unclustered” instead of “homogeneous/heterogeneous”, we find ourselves very much within the currently used nomenclature in the field, as demonstrated by other publications (*Nat. Rev. Mater.* **2019**, 4, 573 – 587, *Nat. Nanotechnol.* **2022**, 17, 190–196, *Matter* **2024**, 7, 1054 – 1070, *Science* **2019**, 363, 627 – 631, *Chem. Mater.* **2016**, 28, 6536 – 6543, *Science* **2022**, 378, 747 - 754). As a result, we have continued using homogeneous/heterogeneous to refer to this behavior in order to be consistent with published works. We have nevertheless now made a clear distinction (based on input from Reviewer #2) between homogeneous/heterogeneous for variations related to composition and luminescence and smooth/rough describe morphological properties.

Page 15, Line 10 – Hyperspectral PL study on composition 0.75/0.60 can be included to make a stronger claim on the statement about an increase in PL heterogeneity with an increase in bromide content. The same study can be included for composition 0.25/0.60 to rule out the possibility of PL heterogeneity increase with Br at lower MA content.

We have considered expanding our compositional space further (75% MA, 60% Br) for local PL measurements. However, we found that these compositions are very sensitive to visible illumination required for the PL measurement such that within the acquisition time, ion migration occurs causing a red-shift in the PL. As a result, measurements that use lower light intensity such as the sensitive EQE measurements (Fig. 6) are more suitable for such compositions.

Fig.6 – In the figure caption the continuous illumination is mentioned as 10 minutes but in the main text it is mentioned as 5 minutes – check this discrepancy

Thank you for pointing out this typo. We have corrected it in the main text. The correct duration is 10 min (see below).

Page 20: “We used continuous illumination (532 nm, 1-Sun equivalent intensity, 10 min.) to induce defect migration and drive halide segregation in the solar cells⁵⁸.”

The claims over increased photostability based on the inclusion of MA without any device or accelerated testing under light + heat is a bit concerning, especially due to the well-characterized instability challenges of MA inclusion. How can the authors be sure that the improvements in morphology from crystallization control with larger MA fractions will not lead to degradation in devices?

Thanks for this interesting question. Indeed, MA has been linked to degradation in previous studies (for example *Energy Environ. Sci.* **2016**, 9, 1655 – 1660). However, the wrinkling behavior we describe in this work is not limited to MA-containing perovskites and has been observed in MA-free perovskites in several prior publications (*Nature Communications* **2021**, 12, 1554, *ACS Energy Letters* **2018**, 3, 6, 1225–1232, *J. Phys. Chem. C* **2018**, 122, 30, 17123–17135, *J. Phys. Chem. Lett.* **2024**, 15, 36, 9255–9262, *ACS Energy Lett.* **2023**, 9, 75–84, *J. Appl. Phys.* **2018**, 123, 175302, *Matter* **2020**, 2, 207–219). The precise role of the cation in this process needs further examination although there is one report in triple-cation (Cs, FA, MA) perovskites that describes Cs heterogeneity in wrinkled films (*J. Phys. Chem. C* **2018**, 122, 23345–23351). It is therefore possible that ionic (cation and halide) heterogeneity also occurs in MA-free materials, similar to what we report in our work. As a result, we speculate that in a MA-free wrinkled film with ionic heterogeneity, the halide migration processes discussed in our work also occur and drive degradation. Therefore, while it is entirely possible that degradation due to MA dissociation is suppressed in MA-free films, degradation due to wrinkling will likely still persist. We have added some text (see below) to the Summary section to highlight the ubiquity of this behavior irrespective of MA content.

Page 22: “We also note previous works that report similar morphological behavior in methylammonium-free perovskite compositions, emphasizing possible limits to charge-carrier dynamics and stability^{11,12,14–16,18}.”

There is a discussion of relevance to tandems and multi-junctions in the work—such as claiming the wrinkled surface is not amenable to multijunction processing—but there are reports of high-efficiency tandems on textured Si with a similar profile to the wrinkled films (<https://www.sciencedirect.com/science/article/pii/S2542435120300350>). Can the authors comment on why they expect this to be different than in the case of Si?

Thank you for raising this interesting issue. Indeed, there are reports of tandem devices that successfully incorporate texturing. Devices using thermal evaporation to coat the perovskite and charge transport layers certainly don't face these processing challenges when depositing on textured surfaces (*Nature Materials* **2018**, 17, 820). There are other reports of solution-processed perovskite sub-cells on top of textured silicon sub-cells with varying degrees of texturing. Here, to our knowledge, only blade coating (*Joule* **2020**, 4, 5, 850) has been reported for coating on top of textures in the range of > 1 μm . The texture heights must be lower for spin-coating to be

applicable (*Nature Nanotechnology* **2022**, 17, 1214 and *Science* **2020**, 367, 6482, 1135). Furthermore, in this work, we are trying to address challenges for devices that are beyond tandem solar cells (triple-junctions and quadruple-junctions, for example), motivating our choice to study perovskite compositions with high bromide contents. There, irrespective of the overall configuration (perovskite-perovskite-silicon or all-perovskite), there is a perovskite-perovskite “sub-tandem” in the stack. To our knowledge, no reports are currently available where a textured perovskite sub-cell has been used in a multijunction device. Finally, another important challenge to consider, in addition to the perovskite deposition, is the conformal coating of a series of thin (< 50 nm), optically benign charge transport and interfacial layers in complex multijunction stacks using solution-processing. And while several layers are deposited using evaporation, sputtering or atomic layer deposition (C₆₀, SnO_x, TCO, anti-reflection coatings), spin-coating and other solution-based methods are common for other layers. For example, commonly used self-assembled monolayers (< 5 nm) are very sensitive to the surface (*Nature* **2023**, 624, 289-294, *ACS Appl. Mater. Interfaces* **2022**, 14, 1, 2166-2176). Solution-processed interfacial layers in devices are also very thin. So, taken together, we strongly believe that overcoming challenges related to wrinkling is extremely relevant to the development of multijunction device stacks where solution-processing is involved in the series of fabrication steps.

The results section has the required details, which are well-elaborated. The hypothesis observed is well-explained and well-validated by relevant tests. However, the results section still needs some refinement to put forth the best representation.

Thank you for the overall positive view of our results. We hope that the changes we have described above have addressed your comments/questions appropriately.

Summary

This section is well-explained.

Thank you for the positive comment.

Materials and Methods

Well-elaborated

Thank you for the positive comment.

RESPONSE TO REVIEWERS

We thank the reviewers for their positive evaluation of our work. We have addressed the comment from Reviewer #1 below. The comment is marked in **BLUE** and the authors' response is marked in **BLACK**.

Reviewer #1 (Remarks to the Author):

I thank the reviewers for addressing my comments. One lingering question remains, though. The authors' explanation of wrinkle formation due to the instability of the elastic/viscoelastic bilayer is reasonable. However, what remains unexplained is the relative richness of the peaks/ridges in iodide and the troughs/valleys in bromide.

We thank the reviewer for their positive evaluation and for this question. The relative richness in iodide of the peaks compared to the valleys in the XRF measurements is related to the overall volume fractions that the iodide-rich (viscoelastic) and iodide-poor (elastic) layers represent in those regions. In the peak-like region, the iodide-rich viscoelastic layer is thicker than in the valley-like region. As a result, in the XRF measurement that probes the bulk composition, the iodide-rich viscoelastic layer represents a larger volume fraction at the peaks and a relatively smaller volume fraction in the valley. As a result, the peaks show an overall richness in iodide compared to the valley.

Reviewer #2 (Remarks to the Author):

The authors have responded to the reviewers' comments and revised the manuscript accordingly. I have no additional remarks.

We thank the reviewer for their positive evaluation.

Reviewer #3 (Remarks to the Author):

The comments and the suggestions were addressed adequately.

We thank the reviewer for their positive evaluation.